# Principled Fast and Meta Knowledge Learners for Continual Reinforcement Learning

**Ke Sun**[1*]**, Hongming Zhang**[2*]**, Jun Jin**[1]**, Chao Gao**[3]**, Xi Chen**[4]**, Wulong Liu**[5]**, Linglong Kong**[1†]

[1]University of Alberta
[2]National Key Laboratory of Cognition and Decision Intelligence for Complex Systems,
Institution of Automation, Chinese Academy of Sciences
[3]Edmonton Research Center, Huawei Canada
[4]Huawei Noah's Ark Lab
[5]BetaInfinity
{ksun6,jjin5,lkong}@ualberta.ca
hongming.zhang@ia.ac.cn
{chao.gao4,xi.chen4}@huawei.com
wulong.liu@gmail.com

## Abstract

Inspired by the human learning and memory system, particularly the interplay between the hippocampus and cerebral cortex, this study proposes a dual-learner framework comprising a fast learner and a meta learner to address continual Reinforcement Learning (RL) problems. These two learners are coupled to perform distinct yet complementary roles: the fast learner focuses on knowledge transfer, while the meta learner ensures knowledge integration. In contrast to traditional multi-task RL approaches that share knowledge through average return maximization, our meta learner incrementally integrates new experiences by explicitly minimizing catastrophic forgetting, thereby supporting efficient cumulative knowledge transfer for the fast learner. To facilitate rapid adaptation in new environments, we introduce an adaptive meta warm-up mechanism that selectively harnesses past knowledge. We conduct experiments in various pixel-based and continuous control benchmarks, revealing the superior performance of continual learning for our proposed dual-learner approach relative to baseline methods. The code is released in https://github.com/datake/FAME.

## 1 Introduction

Most deep reinforcement learning (RL) algorithms (Sutton & Barto, 2018; Mnih et al., 2015; Schulman et al., 2015; Haarnoja et al., 2018a; Schulman et al., 2017) are designed for a single task, where the environmental dynamics and reward function often remain stationary over time. In contrast, humans continually face diverse and evolving environments, learning to solve new tasks sequentially throughout their lives. Building artificial general agents with such adaptive capabilities requires continual learning, i.e., the ability to acquire new knowledge efficiently without forgetting previously learned skills. In this realm, *Continual Reinforcement Learning* (Khetarpal et al., 2022; Abel et al., 2023) emerges as a crucial paradigm, aiming to balance *plasticity* (rapid adaptation to new tasks) and *stability* (retention of past knowledge). An ideal continual learning agent is capable of transferring useful knowledge forward to accelerate learning in new environments while avoiding *catastrophic forgetting* (Dohare et al., 2024) across previously encountered tasks.

Recent work in continual RL spans a wide range of strategies (Barreto et al., 2020; Kessler et al., 2022; Kaplanis et al., 2019; Caccia et al., 2022; Kaplanis et al., 2018; Gaya et al., 2023; Yang et al., 2023; Wolczyk et al., 2022; Anand & Precup, 2023; Wan et al., 2022; Chandak et al., 2020; Sun et al., 2025; Liu et al., 2025; Tang et al., 2025). Despite rapid progress, continual RL remains fragmented: existing algorithms above are often motivated by heuristics or developed from distinct

---

* Equal Contribution.
† Corresponding Author.

perspectives (see Appendix A for a detailed comparison of related work), but there is no principled way to understand *when* and *why* they work. This lack of a theoretical foundation makes it difficult to assess when knowledge transfer will be beneficial, how to mitigate forgetting, and how to set explicit optimization objectives, hindering principled algorithm development in continual RL.

To address these challenges, our study contributes to new foundations of continual RL by (1) defining the MDP difference to formally quantify environment similarity, and (2) introducing a quantitative measure of catastrophic forgetting applicable to both value- and policy-based RL. Building on these new foundations and principles, we propose a dual-learner paradigm that mirrors hippocampal–cortical interactions in the human learning and memory systems (Kumaran et al., 2016), decomposing continual RL into two complementary objectives: *knowledge transfer* and *knowledge integration*. Specifically, we maintain two distinct yet complementary components, i.e., a fast learner and a meta learner, which are analogous to the functional roles of the hippocampus (the fast learner) and the neocortex (the meta learner) in the human brain. Our approach also elucidates its more profound connection to the transfer and multi-task RL problems (Schaul et al., 2015; Rusu et al., 2015; Parisotto et al., 2016; Rajendran et al., 2017; Teh et al., 2017; Bai et al., 2023).

**Knowledge Transfer via Fast Learner.** We leverage a fast learner to rapidly acquire knowledge from a new task by adaptively transferring prior knowledge stored in a meta learner. To circumvent the potential negative transfer issue, an *adaptive meta warm-up* strategy is developed by either directly copying parameters as initialization or adding a behavior cloning regularization in the early training phase to guide the exploration. The function of a fast learner in knowledge transfer resembles the hippocampus. By swiftly encoding the new experiences and discriminating the effectiveness of existing knowledge, the hippocampus, guided by the neocortex, specifically functions to quickly assimilate novel scenarios in response to immediate environmental changes or drifts.

**Knowledge Integration via Meta Learner.** After assimilating the new knowledge by the fast learner, an incremental knowledge integration incorporates the new experiences into the existing knowledge pool stored in the meta learner. Under the new foundation, the knowledge integration is incrementally updated in the principle of *catastrophic forgetting minimization* under specific divergence metrics. After consolidating old and new experiences, the meta learner enhances the adaptive meta warm-up, facilitating the knowledge transfer in the next environment. The knowledge integration process plays a role akin to the cerebral cortex, which gradually integrates, incorporates, and consolidates new knowledge into the existing cognitive structure in the human brain to build a more generalizable, robust, and stable decision-making system.

**Contributions.** The contributions of our study can be succinctly summarized as follows:

- We propose new foundations of continual RL, including the definition of MDP difference and the measure of catastrophic forgetting, underpinning the algorithmic innovations in the future.

- We devise a dual-learner system that incorporates distinct yet complementary fast and meta learners to perform knowledge transfer and knowledge integration. The interplay between fast and meta learners mimics the hippocampal-cortical dialogue in the brain's memory systems.

- We provide comprehensive empirical studies to validate the efficiency of our dual-learner system in discrete and continuous action domains, across both value- and policy-based RL algorithms.

## 2 PROBLEM SETTING AND NEW CONTINUAL RL FOUNDATIONS

**Problem Setting.** Let $[K]$ denote $\{1, 2, ..., K\}$. We consider a sequence of $K$ tasks, where each task $k \in [K]$ is modeled by a Markov Decision Process (MDP) $\mathcal{M}_k = \langle \mathcal{S}_k, \mathcal{A}_k, P_k, R_k, \gamma \rangle$. $\mathcal{S}_k$ and $\mathcal{A}_k$ denote the state and action spaces, $P_k : \mathcal{S}_k \times \mathcal{A}_k \to \mathcal{P}(\mathcal{S}_k)$ is the transition dynamics, $R_k : \mathcal{S}_k \times \mathcal{A}_k \to \mathbb{R}$ is the reward function, and $\gamma$ is the discounting factor. We define the action-value function $Q^\pi(s, a) = \mathbb{E}_\pi \left[ \sum_{i=0}^{\infty} \gamma^i R_{t+i+1} \mid S_t = s, A_t = a \right]$ given a state $s$, an action $a$, and a policy $\pi$. Following common practice in continual RL (Wolczyk et al., 2022; Khetarpal et al., 2022; Malagon et al., 2024), we adopt three assumptions: (1) the same state and action spaces, (2) known task boundaries, i.e., semi-continual RL (Anand & Precup, 2023), and (3) a training budget with a moderate model size and an allowable computation cost. An "optimal" policy can generalize favorably across all tasks by balancing adaptation to new tasks with prior knowledge retention.

**Foundation 1: MDP Distance.** Theoretical analysis in continual RL requires a quantitative similarity measure between environments to determine when knowledge transfer will be beneficial or harmful, and to assess how strongly new tasks may interfere with previously learned ones. A desirable MDP distance should consider variations from both reward functions and transition dynamics between MDPs. To this end, we utilize the distance between two MDP-determined optimal Q functions or task-specific optimal policies to quantify the MDP distance in Definition 1.

**Definition 1.** *(MDP Distance) For two finite MDPs: $MDP_1 = (\mathcal{S}, \mathcal{A}, R_1, P_1, \gamma)$ and $MDP_2 = (\mathcal{S}, \mathcal{A}, R_2, P_2, \gamma)$, we denote their optimal Q functions as $Q_1^*$ and $Q_2^*$ and the optimal policies as $\pi_1^*$ and $\pi_2^*$. The Q-value-based and policy-based MDP distances are defined as $d_Q(Q_1^*, Q_2^*)$ and $d_\pi(\pi_1^*, \pi_2^*)$ under certain divergences or distances $d_Q$ and $d_\pi$, e.g., the $\ell_2$ loss or the KL divergence.*

**Foundation 2: Catastrophic Forgetting.** Our definition of catastrophic forgetting in continual RL is inspired by *distribution drift* and *catastrophic forgetting* quantified in deep learning literature (Doan et al., 2021), which we briefly recap in Appendix B. Grounded in the definition of MDP difference in Definition 1, we introduce catastrophic forgetting between two MDPs in Definition 2. Denote $\mu_k^\pi$ and $\mu_k^Q$ as the state visitation distributions when a policy $\pi$ or a greedy policy $\pi^Q$ over $Q$ defined by $\pi^Q(\cdot|s) = \arg\max_a Q(s, a)$, interacts in the $k$-th environment in a sequence of $K$ tasks.

**Definition 2.** *(Catastrophic Forgetting across Two Environments) Denote $Q_{k-1}, Q_k$ and $\pi_{k-1}, \pi_k$ as Q functions and policies after training RL algorithms across the $(k-1)$-th and $k$-th environments sequentially. The catastrophic forgetting, denoted by CF, is defined as*

$$CF(Q_{k-1}, Q_k) = \sum_{s,a} \mu_{k-1}^{Q_{k-1}}(s)\pi^{Q_{k-1}}(a|s)d_Q\left(Q_{k-1}(s,a), Q_k(s,a)\right), \tag{1}$$

$$CF(\pi_{k-1}, \pi_k) = \sum_s \mu_{k-1}^{\pi_{k-1}}(s)d_\pi\left(\pi_k(\cdot|s), \pi_{k-1}(\cdot|s)\right). \tag{2}$$

For each $s$ and $a$, the weights $\mu_{k-1}^{Q_{k-1}}(s)\pi^{Q_{k-1}}(a|s)$ and $\mu_{k-1}^{\pi_{k-1}}(s)$ characterize the *relative importance* when measuring discrepancies between Q functions and policies. Crucially, we evaluate these weights using the preceding policy $\pi_{k-1}$ ($\pi^{Q_{k-1}}$) rather than the current policy $\pi_k$ ($\pi^{Q_k}$), as the past policy better reflects states and actions that mattered most in the old task. In contrast, if we use $\pi_k$ ($\pi^{Q_k}$) for the weight evaluation, significant changes in the Q-function or policy on previously important state–action pairs might be overlooked, since the new policy may no longer visit them.

## 3  FAME: Principled FAst and MEta Knowledge Continual RL

The proposed FAME approach is applicable to both value-based and policy-based RL. In value-based RL, we denote $Q_k$ as the updated fast learner after learning task $k$, followed by a meta learner $Q_k^M$ that integrates knowledge from the preceding meta learner $Q_{k-1}^M$ and $Q_k$. In policy-based RL, we denote $\pi_k$ as the fast learner after learning task $k$, and then a meta learner $\pi_k^M$ integrates knowledge from the preceding meta learner $\pi_{k-1}^M$ and $\pi_k$. The coupled updating of the fast and meta learners in FAME is illustrated in Figure 1. In principle, the fast learner rapidly learns the new task guided by the meta learner

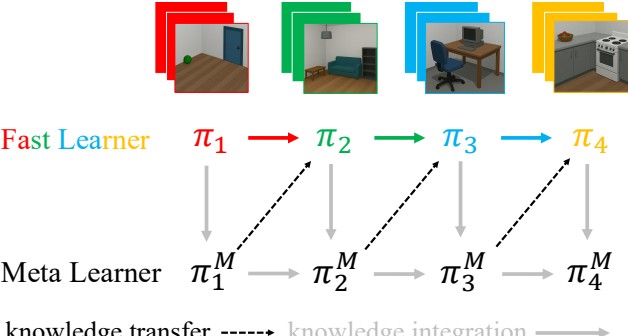

Figure 1: Illustration of FAME. In value-based continual RL, the fast learner can be denoted by $\{Q_k\}_{k=1}^K$ accordingly instead of $\{\pi_k\}_{k=1}^K$.

via the proposed adaptive meta warm-up. Meanwhile, the meta learner consolidates the experience from the preceding meta learner and the current fast learner via knowledge integration to minimize catastrophic forgetting.

### 3.1 VALUE-BASED CONTINUAL RL WITH DISCRETE ACTION SPACES

#### 3.1.1 KNOWLEDGE INTEGRATION: CATASTROPHIC FORGETTING MINIMIZATION PRINCIPLE

After the fast learner $Q_k$ completes training in the $k$-th environment, the knowledge integration phase begins. In this phase, the meta learner $Q_k^M$ is updated to consolidate information by combining the prior knowledge encoded in the preceding meta learner $Q_{k-1}^M$ and the new knowledge acquired by the fast learner $Q_k$. Unlike classical multi-task RL, which maximizes the average rewards, our meta learner is explicitly designed to minimize the catastrophic forgetting defined in Definition 2.

**Q-Value-based Catastrophic Forgetting and Limitations.** We extend the Q-value-based catastrophic forgetting defined in Eq. 1 over a sequence of $K$ tasks. In the $k$-th environment, the optimal meta Q value function $Q_k^M$ is the minimizer by solving the following objective function:

$$Q_k^M = \arg\min_{\widetilde{Q}_k^M} \sum_{i=1}^{k} \sum_{s,a} \mu_i^{Q_i}(s)\pi^{Q_i}(a|s) \left(Q_i(s,a) - \widetilde{Q}_k^M(s,a)\right)^2, \tag{3}$$

where we recall that $\mu_i^{Q_i}$ is the state visitation distribution when the greedy policy $\pi^{Q_i}$ (i.e., $\pi^{Q_i}(\cdot|s) = \arg\max_a Q_i(s,a)$) interacts with the $i$-th environment. Intuitively, the direct minimizer $Q_k^M$ by solving Eq. 3 is a weighted average among $\{Q_i\}_{i=1}^k$. However, developing the capability of continual learning by storing all previous Q functions fails to scale in the number of tasks, which is one of the crucial requirements in continual RL. Indeed, we can rewrite the above objective function as an incremental updating rule between the preceding meta learner $Q_{k-1}^M$ and the fast learner $Q_k$, which we defer to Appendix C.1. Nonetheless, the fundamental limitation of Q-value-based catastrophic forgetting lies in the fact that it is mainly applicable to distinct environments with **similar scales of Q values**, such as those with varying transition dynamics yet the same reward function. As the new arriving environment is agnostic, the scale of the Q-values may be hard to learn because it is not necessarily bounded and can be quite unstable (Rusu et al., 2015). The previously well-learned tasks with high rewards tend to be more salient in consolidating knowledge than those with small rewards (Zhang et al., 2023). Therefore, the policy-based definition of catastrophic forgetting in Eq. 2 is more versatile than the Q-value-based one in Eq. 1, serving as a preferable alternative. In addition, policies may inherently enjoy lower variance than value functions, contributing to improved performance and stability (Greensmith et al., 2004).

**Policy-based Catastrophic Forgetting.** Even in value-based continual RL, it is preferable to employ the policy-based definition of catastrophic forgetting based on Eq. 2 for an incremental update of the meta learner. Akin to the Q-value-based catastrophic forgetting in Eq. 3, the optimal meta policy $\pi_k^M$ in the $k$-th environment is the minimizer by solving the following objective function:

$$\pi_k^M = \arg\min_{\widetilde{\pi}_k^M} \sum_{i=1}^{k} \sum_{s} \mu_i^{\pi_i}(s)d_\pi\left(\pi_i(\cdot|s), \widetilde{\pi}_k^M(\cdot|s)\right). \tag{4}$$

**Incremental Softmax Meta Learner Update for Value-based Continual RL.** Define the weight function $w_i^Q(s,a) = \mu_i^{Q_i}(s)\pi^{Q_i}(a|s)$ for each $i \in [K]$. For any measurable function $f(s,a)$ and weight function $w$ with $\sum_{s,a} w(s,a) = 1$ and $w(s,a) \geq 0$ for each $s$ and $a$, we define $\mathbb{E}_w[f] = \sum_{s,a} w(s,a)f(s,a)$. When equipped with the categorical representation, the Q-values can be converted into a Softmax (Boltzmann) policy, allowing the value-based continual RL to minimize the policy-based catastrophic forgetting objective defined in Eq. 4. Specifically, given a temperature $\tau$, we denote $\pi^{Q_i}(a|s) = \exp\left(Q_i(a|s)/\tau\right) / \sum_{a'} \exp\left(Q_i(a'|s)/\tau\right)$. By employing the KL divergence as $d_\pi$, we derive a concise meta learner update rule in Proposition 1.

**Proposition 1** (Incremental Softmax Q-Value-based Meta Learner Update). *Denote* $\widetilde{\pi}_k^M(a|s) = \exp\left(\widetilde{Q}_k^M(a|s)/\tau\right) / \sum_{a'} \exp\left(\widetilde{Q}_k^M(a'|s)/\tau\right)$. *After a softmax policy transformation, the Q-value-based meta learner incremental update is written as*

$$Q_k^M = \arg\min_{\widetilde{Q}_k^M} \sum_{i=1}^{k-1} \mathbb{E}_{w_i^Q}\left[\log\frac{\pi_{k-1}^M}{\widetilde{\pi}_k^M}\right] + \mathbb{E}_{w_k^Q}\left[\log\frac{\pi^{Q_k}}{\widetilde{\pi}_k^M}\right] = \arg\max_{\widetilde{Q}_k^M} \sum_{i=1}^{k} \mathbb{E}_{w_i^Q}\left[\log\widetilde{\pi}_k^M\right]. \tag{5}$$

The proof of Proposition 1 is straightforward and is therefore deferred to Appendix C.2 for completeness. Crucially, minimizing the policy-based catastrophic forgetting in Eq. 5 is simply equivalent to the Maximum Likelihood Estimator (MLE) by fitting the meta learner $Q_k^M$ to a mixture of state-action distributions across encountered environments. We highlight that the final simplified objective in Eq. 5 is independent of $Q_{k-1}^M$ and $Q_k$; however, knowledge integration in principle takes the form of an incremental update rule, which we instantiate in Section 3.2.1.

### 3.1.2 KNOWLEDGE TRANSFER VIA ADAPTIVE META WARM-UP

**Challenges.** An effective knowledge transfer necessitates rapidly adapting to the new environment by taking advantage of the previous knowledge if accessible. However, the commonly used finetuning is effective when tasks are similar, but lead to *negative transfer* issue that frequently occurs in continual RL (Ahn et al., 2025; Wolczyk et al., 2022). The negative transfer, a crucial factor of the *loss of plasticity* (Dohare et al., 2024), leads to performance degradation owing to the dissimilarity between the two tasks. Training from scratch (i.e., reset) is easy to implement to circumvent the negative transfer (Chen et al., 2024; Ahn et al., 2025). However, this naive warm-up lacks flexibility and fails to make full use of the accumulated knowledge to speed up the adaptation to a new task.

**Adaptive Meta Warm-Up via One-vs-all Hypothesis Test.** When a new task arrives, it is a common strategy to initialize the fast learner with parameters from the meta learner. Nonetheless, knowledge and skills previously acquired in past tasks may become misleading when the new environment differs substantially. For instance, it is particularly evident that humans make incorrect decisions or take suboptimal actions when the new information contradicts earlier experiences. To harmonize the potentially conflicting objectives, we propose an *adaptive meta warm-up approach* that chooses the most effective warm-up strategy among **the preceding meta learner, a random learner (i.e., reset), and the preceding fast learner (i.e., finetune)**. Formally, the adaptive meta warm-up is framed as a one-vs-all hypothesis test based on policy evaluation during the early interaction with a new environment. When the $k$-th task arrives, we have access to three types of warm-up learners, including a preceding fast learner $Q_{k-1}$, a meta policy $\pi_{k-1}^M$ with the softmax transformation from $Q_{k-1}^M$, and a random Q function $Q^0$ associated with the policy $\pi^0$. Evaluating the three warm-up learners yields their value functions defined by $V_k^f = \mathbb{E}_{\pi_{k-1}}[R]$, $V_k^M = \mathbb{E}_{\pi_{k-1}^M}[R]$, and $V_k^r = \mathbb{E}_{\pi^0}[R]$. For each task $k$ that arrives, the one-vs-all hypothesis test with a composite null is expressed as

$$H_0 : V_k^M \leq \max\left\{V_k^f, V_k^r\right\} \quad \text{vs.} \quad H_1 : V_k^M > \max\left\{V_k^f, V_k^r\right\}. \tag{6}$$

When the null hypothesis $H_0$ cannot be rejected, we further compare $V_k^f$ and $V_k^r$ via a common parametric hypothesis test, e.g., $t$-test. In most scenarios, picking the best warm-up strategy according to the empirical ranking often performs favorably. However, in safety-critical scenarios, e.g., autonomous driving, a rigorous statistical test is crucial either by bootstrapping or anytime valid inference (Ramdas et al., 2023) on the adaptively collected dataset used for the policy evaluation.

**Meta Warm-Up via Behavior Cloning Regularization.** Once we reject $H_0$, we are ready to perform the meta warm-up. However, directly initializing the fast learner $Q_k$ via the meta policy $\pi_{k-1}^M$ is infeasible as the meta learner is now represented as a policy instead of a Q function under the update in Proposition 1. An easy and effective way to address this policy-to-value transfer mismatch is to impose Behavior Cloning (BC) regularization in the early training phase, when the meta policy $\pi_k^M$ serves as the expert for data collection and early exploration. Concretely, $Q_k$ is the minimizer of the BC regularized loss $L(Q_k) = L_0(Q_k) + \lambda \mathbb{E}_s \left[ \text{KL}(\pi_{k-1}^M(\cdot|s) || \pi^{Q_k}(\cdot|s)) \right]$, where $L_0(Q_k)$ is the original loss to update $Q_k$, such as the MSE or Huber loss in DQN (Mnih et al., 2015).

### 3.1.3 ALGORITHM: VALUE-BASED FAME

**Meta Buffer $\mathcal{M}$ in Knowledge Integration.** In the *last $N$* steps of updating the fast learner in each environment, we additionally store the state-action pairs in a meta learner's buffer $\mathcal{M}$, which are used to approximate $w_i^Q$ for $i \in [k]$ in Eq. 5. Note that the stored state-action pairs are only a small portion of the training dataset for each task (around $1\%$ or $2\%$ in our experiments), yielding a moderate size of the meta buffer $\mathcal{M}$. The moderate and fixed size of a meta buffer is crucial in reality, as we are not expected to store too much past data in continual RL.

---

**Algorithm 1** Value-based FAME Update in the $k$-th Environment

---

1: **Initialize**: Fast Buffer $\mathcal{F}$, Meta Buffer $\mathcal{M}$, $Q_{k-1}^M$, $Q_{k-1}$, $Q^0$, Warm-Up Step $L$, Estimation Step $N$.
2: # Knowledge Transfer: Adaptive Meta Warm-Up
3: Initialize $Q_k$ in $\{Q_{k-1}, Q_k^M, Q^0\}$ via Eq. 6 within $L$ steps
4: **for** $t = L$ to $T$ **do**
5:    Observe $S_t$, take action $A_t$, receive $R_t$, observe $S_{t+1}$
6:    Store $(S_t, A_t, R_t, S_{t+1})$ in $\mathcal{F}$
7:    Update $Q_k$
8:    **if** $t > T - N$ **then**
9:       Store $(S_t, A_t)$ in $\mathcal{M}$ # To Estimate $w_k^Q$
10:    **end if**
11: **end for**
12: Reset $\mathcal{F}$
13: # Knowledge Integration: Minimize Catastrophic Forgetting
14: Update $Q_k^M$ via Eq. 5 using state-action pairs in $\mathcal{M}$

---

**Algorithm.** Denote the buffer of the fast learner as $\mathcal{F}$ and $Q^0$ as the randomly initialized Q function. We denote $T$ as the timesteps in each environment. As suggested in Algorithm 1, when the $k$-th environment arrives, we warm start the fast learner $Q_k$ via the adaptive meta warm-up strategy among the preceding meta learner $Q_{k-1}^M$, the preceding fast learner $Q_{k-1}$ and a random learner $Q^0$ (i.e., reset) within the first $L$ steps. The adaptive meta warm-up makes full use of previous information to perform an adaptive knowledge transfer. Once the $k$-th task ends, the knowledge integration phase starts, when the meta learner $Q_k^M$ is updated via Eq. 5 on the data collected in the meta buffer $\mathcal{M}$. The meta learner $Q_k^M$ incorporates the acquired knowledge in $Q_k$ into $Q_{k-1}^M$ via an incremental update rule in principle.

### 3.2 POLICY-BASED CONTINUAL RL WITH CONTINUOUS ACTION SPACES

#### 3.2.1 KNOWLEDGE INTEGRATION: CATASTROPHIC FORGETTING MINIMIZATION PRINCIPLE

As opposed to the value-based continual RL with an incremental softmax meta learner updates in Proposition 1, in policy-based continual RL, we directly minimize the policy-based catastrophic forgetting in Eq. 4 regarding the parameterized policy function. The detailed incremental update rule depends on **the choice of $d_\pi$ and how we represent the policy** in a continuous action space. Next, we will introduce two variants of policy-based continual RL methods when equipped with the forward KL divergence and Wasserstein distance, respectively.

**Method 1 (FAME-KL): Policy Distillation under Forward KL Divergence.** We show that the policy-based knowledge integration reduces to a form of policy distillation. Analogous to Proposition 1 in the value-based continual RL, the policy-based knowledge integration in Eq. 4, when instantiated with the forward KL divergence under an accessible probabilistic policy, yields an update rule as

$$\pi_k^M = \arg\max_{\widetilde{\pi}_k^M} \sum_{i=1}^{k} \mathbb{E}_{w_i} \left[ \log \widetilde{\pi}_k^M \right], \tag{7}$$

where we recall that $w_i(s, a) = \mu_i^{\pi_i}(s)\pi_i(a|s)$ denotes the policy-based steady state-action distribution on the $i$-th environment. Importantly, the policy-based knowledge integration objective above coincides with the knowledge distillation update used in policy distillation (Rusu et al., 2015) and with typical multi-task RL formulations (Teh et al., 2017), establishing an intriguing connection.

**Method 2 (FAME-WD): Wasserstein Distance (WD)-based Knowledge Integration.** Although the KL divergence is often adopted for its computational simplicity, the Wasserstein distance is preferable when comparing more complex policy distributions. In contrast to the KL divergence, which only measures pointwise differences between distributions, the Wasserstein distance by definition explicitly accounts for the underlying geometry of the data space, such as the probability space of the policy outputs, and therefore provides a more informative notion of distributional discrepancy (Panaretos & Zemel, 2019; Arjovsky et al., 2017). In Proposition 2, we derive the policy-based

incremental update rule under the Wasserstein distance. Particularly, when the policy function is represented by a (multivariate) Gaussian distribution, as is common in many policy-based algorithms, a closed-form incremental update rule becomes available, enabling efficient knowledge integration for the meta learner. The proof is given in Appendix C.3.

**Proposition 2** (Incremental Policy-based Meta Learner Update under Wasserstein Distance)**.** *Consider $d_\pi$ to be the squared 2-Wasserstein distance denoted by $W_2^2$ in Eq. 2 of Definition 2. The policy is represented as an independent (multivariate) Gaussian distribution over the action $a$. Minimizing policy-based catastrophic forgetting in Eq. 4 is equivalent to:*

$$\pi_k^M = \arg\min_{\widetilde{\pi}_k^M} \left\{ \sum_{i=1}^{k-1} \sum_s \mu_i^{\pi_i}(s) W_2^2 \left( \widetilde{\pi}_k^M(\cdot|s), \pi_{k-1}^M(\cdot|s) \right) + \sum_s \mu_k^{\pi_k}(s) W_2^2 \left( \widetilde{\pi}_k^M(\cdot|s), \pi_k(\cdot|s) \right) \right\}. \quad (8)$$

### 3.2.2 KNOWLEDGE TRANSFER VIA ADAPTIVE META WARM-UP

For the adaptive meta warm-up in policy-based RL, we perform policy evaluation across the first $L$ steps and conduct the one-vs-all hypothesis test in Eq. 6 when a new task arrives, which is similar to value-based continual RL. Once we determine the best-performing warm-up policy among the fast policy $\pi_{k-1}$, the meta policy $\pi_{k-1}^M$, and a random policy $\pi^0$, we directly initialize the fast policy. Using parameter initialization as the meta warm-up strategy is more convenient for deployment than adding the BC regularization used in the value-based continual RL introduced in Section 3.1.2.

**Algorithm.** Since the description of the policy-based FAME algorithm closely parallels Algorithm 1, we defer it with a discussion of computational cost and practical guidance to Appendix D. We also provide a general discussion of FAME framework in Appendix E.

## 4 EXPERIMENTS

In this section, we validate our FAME approach across a sequence of tasks from multiple environments and domains, including the pixel-based tasks with a discrete action space in Section 4.1 and control problems with a continuous action space in Section 4.2. The central hypothesis is that the interplay between knowledge transfer and knowledge integration of the fast and meta learners in FAME benefits both forward transfer (i.e., plasticity) and catastrophic forgetting (i.e., stability).

**Evaluation Metrics.** We employ the standard metrics (Wolczyk et al., 2021; 2022) in continual RL to evaluate *average performance*, *forgetting* to measure stability, and *forward transfer* to quantify plasticity. Let $p_i(t)$ be the success rate or average returns in task $i$ by using the policy at time $t$ with $t \in [K \cdot T]$, where $K$ is the number of environments, and $T$ is the total timesteps in each task. $p_i(t)$ is task-specific with $p_i(t) \in \mathbb{R}$ for our pixel-based tasks and $p_i(t) \in [0, 1]$ in our control tasks.

- **Average Performance.** The average performance is evaluated on the policy at time $t$ across all $K$ tasks by $P_K(t) = \frac{1}{K} \sum_{i=1}^{K} p_i(t)$. By default, the average performance is calculated on the final policy when $t = K \times T$. For FAME, this metric is calculated on the meta learner.

- **Forward Transfer (FT)**: The forward transfer is defined as the normalized area between the training curve of the considered algorithm and the baseline. Namely, FT $= \frac{1}{K} \sum_{i=1}^{K} \text{FTr}_i$ with

$$\text{FTr}_i = \frac{\text{AUC}_i - \text{AUC}_i^b}{1 - \text{AUC}_i^b}, \quad \text{AUC}_i = \frac{1}{\Delta} \int_{(i-1)\cdot\Delta}^{i\cdot\Delta} p_i(t) \mathrm{d}t, \quad \text{AUC}_i^b = \frac{1}{\Delta} \int_{(i-1)\Delta}^{i\Delta} p_i^b(t) \mathrm{d}t. \quad (9)$$

To evaluate this metric in pixel-based tasks, we first normalize $p_i(t)$ in each task to ensure $\text{AUC}_i \in [0, 1]$ and then we calculate a normalized metric of the forward transfer.

- **Forgetting (F)**: Forgetting is the performance difference between the policy at the end of a task and after the whole sequence of tasks. Namely, $F = \frac{1}{K} \sum_{i=1}^{K} F_i$ with $F_i = p_i(i \cdot T) - p_i(K \cdot T)$.

**Experimental Setup. (1)** For the pixel-based tasks, we perform experiments on MinAtar and Atari games (Young & Tian, 2019; Bellemare et al., 2013). MinAtar is a commonly used continual RL benchmark (Anand & Precup, 2023; Tang et al., 2025) with relatively lighter computational requirements, allowing us to sweep a range of hyperparameters, report statistical results averaged over 30

seeds, and probe the mechanism and advantages of our proposal. We employ DQN (Mnih et al., 2015) in breakout, freeway, and spaceinvaders games, and run for 3.5M steps by randomly choosing each of the three games every 500k steps, i.e., 7 tasks in each sequence. We study two sequences of Atari games following (Malagon et al., 2024), with the 10 playing modes of the *ALE/SpaceInvaders-v5* environment and 7 playing modes of the *ALE/Freeway-v5* environment. We run Proximal Policy Optimization (PPO) (Schulman et al., 2017) and each task is run for $1M$ timesteps. **(2)** For the robotics arm manipulation tasks, we employ Meta-World (Yu et al., 2020), a standard and established benchmark commonly employed in continual RL (Malagon et al., 2024; Chung et al., 2024; Ahn et al., 2025). Note that the traditional Continual World (Wolczyk et al., 2021) is built on top of Meta-World with a specific sequence of tasks (e.g., CW10 or CW20). Instead, we evaluate on 3 randomly selected task sequences following (Chung et al., 2024) in Meta-World (see Appendix G.1), offering a more flexible and robust evaluation. We deploy the Soft Actor-Critic (SAC) algorithm (Zhang et al., 2020; Haarnoja et al., 2018a) with $1M$ timesteps on each task with 10 tasks in each sequence.

## 4.1 Pixel-based Environments with Discrete Action Spaces

**Comparison Methods.** **(1)** For MinAtar, we follow (Anand & Precup, 2023) and compare our `FAME` approach with DQN (`Reset`), DQN-Finetune (`Finetune`), DQN with a large buffer (`LargeBuffer`), DQN with multi-heads that knows the task identity (`MultiHead`), `PT-DQN` (Anand & Precup, 2023). Both fast and meta learners in our `FAME` method employ the same DQN architecture. (2) For Atari games, we add `PackNet` (Mallya & Lazebnik, 2018) and `ProgressiveNet` (Rusu et al., 2016) as baselines. Both fast and meta learners in FAME adopt the same PPO architecture. Except for `Finetune`, we reset the parameters of all baseline methods when each new environment arrives. By contrast, `FAME` applies the adaptive meta warm-up among fast, random initialization, and initial learning with behavior cloning regularization in Section 3.1.2. More details of our experimental setup and hyperparameters are given in Appendix F.1.

Table 1: Main results on **MinAtar** on Average Performance (*Avg. Perf*), Forward Transfer (*FT*), and Forgetting. Results (Mean $\pm$ SE) are averaged over 10 sequences, each with 3 seeds. $\uparrow$ denotes a positive metric (more is better), while $\downarrow$ is a negative one (less is better). `Reset` is the baseline for evaluating FT. Forgetting is normalized by the standard deviation in each task.

| Method | Ave. Perf $\uparrow$ | | | FT $\uparrow$ | Forgetting $\downarrow$ |
| --- | --- | --- | --- | --- | --- |
| | Breakout | Spaceinvader | Freeway | | |
| Reset | $6.51 \pm 1.67$ | $3.29 \pm 3.09$ | $0.74 \pm 0.38$ | $0.00 \pm 0.00$ | $1.31 \pm 0.23$ |
| Finetune | $10.62 \pm 2.75$ | $4.95 \pm 2.92$ | $0.89 \pm 0.49$ | $0.13 \pm 0.03$ | $1.26 \pm 0.32$ |
| MultiHead | $6.85 \pm 1.76$ | $3.26 \pm 2.99$ | $0.94 \pm 0.42$ | $-0.01 \pm 0.00$ | $1.25 \pm 0.22$ |
| LargeBuffer | $10.71 \pm 2.84$ | $3.24 \pm 2.91$ | $1.16 \pm 0.59$ | $\mathbf{0.16 \pm 0.02}$ | $1.65 \pm 0.33$ |
| PT-DQN | $0.39 \pm 0.02$ | $0.00 \pm 0.00$ | $0.00 \pm 0.00$ | $0.07 \pm 0.02$ | $1.64 \pm 0.02$ |
| FAME | $\mathbf{14.54 \pm 0.58}$ | $\mathbf{18.72 \pm 0.52}$ | $\mathbf{1.69 \pm 0.17}$ | $\mathbf{0.16 \pm 0.03}$ | $\mathbf{0.72 \pm 0.13}$ |

**Main Results: MinAtar.** Table 1 summarizes the metric scores of all methods, demonstrating that `FAME` consistently outperforms other baselines in improving knowledge transfer and retaining all knowledge to mitigate catastrophic forgetting. Notably, for the average performance, `FAME` is most stable with minimal variations among all algorithms except for `PT-DQN`, for which the *permanent value function* (i.e., the counterpart of the meta learner) in (Anand & Precup, 2023) has limited capability to retain the knowledge and thus keeps almost zero average performance. Regarding the forward transfer, `LargeBuffer` performs similarly to `FAME` as storing more past knowledge also contributes to adapting to a known environment. A detailed sensitivity analysis about $\lambda$, Warm-Up step $L$, and Estimation step $N$ is provided in Appendix F.2.

**Performance of Knowledge Integration.** Figure 2 (left) presents the average performance of all methods at the end of each task, reflecting the tendency of catastrophic forgetting. It turns out that `FAME` achieves the highest average performance in the whole training process in most cases, validating the effectiveness of the meta learner in retaining information through knowledge integration.

**Performance of Adaptive Meta Warm-Up in Knowledge Transfer.** Figure 2 (right) exhibits the warm-up selection ratio when the agent encounters different types of arriving environments, revealing the adaptive mechanism. Concretely, if the agent has already stored relevant data previously in $\mathcal{M}$ about the arriving environment, the meta warm-up is chosen with a $95.1\%$ probability. When a

new task occurs against the agent's knowledge, the random initialization is more commonly selected in the adaptive meta warm-up. The learning curves of all algorithms are given in Appendix F.3.

**Main Results: Atari games.** We further compare `FAME` with more baselines on two sequences of Atari games: *ALE/SpaceInvaders-v5* and *ALE/Freeway-v5*. Table 2 showcases that `FAME` outperforms all baselines in terms of average performance and forward transfer. While `Finetune` also benefits from forward transfer, especially on SpaceInvader, where its forward transfer is comparable to `FAME`, we hypothesize that this advantage arises from the similar underlying environmental dynamics and objectives in

Table 2: Main results on **Atari games** on Average Performance (*Avg. Perf*) and Forward Transfer (*FT*). Results (Mean ± SE) are averaged over 3 seeds. The Forgetting metric is omitted as `PackNet` and `ProgressiveNet` store past model parameters and have zero forgetting. `Reset` is the baseline for FT.

| Method | Freeway | | SpaceInvader | |
|---|---|---|---|---|
| | Avg. Perf ↑ | FT ↑ | Avg. Perf ↑ | FT ↑ |
| Reset | $0.16 \pm 0.18$ | 0.00 | $0.10 \pm 0.22$ | 0.00 |
| Finetune | $0.21 \pm 0.17$ | 0.53 | $0.61 \pm 0.41$ | **0.65** |
| ProgressiveNet | $0.39 \pm 0.25$ | 0.21 | $0.61 \pm 0.03$ | 0.06 |
| PackNet | $0.41 \pm 0.24$ | 0.18 | $0.47 \pm 0.06$ | 0.17 |
| FAME | $\mathbf{0.90 \pm 0.12}$ | **0.68** | $\mathbf{0.96 \pm 0.02}$ | 0.63 |

each sequence of tasks, despite the distinct playing modes. We also provide the learning curves of all considered algorithms in Appendix F.3.

## 4.2 ROBOTIC MANIPULATION TASKS WITH CONTINUOUS ACTION SPACES

**Comparison Methods.** (1) `Reset`; (2) `FineTune`; (3) `Average`: we average the Temporal Difference (TD) targets among all past tasks in evaluating the critic loss; (4) `PackNet`; (5) `FAME-KL`: we employ knowledge integration under KL in Eq. 7 (Method 1); (6) `FAME-WD`: we apply knowledge integration under Wasserstein distance in Eq. 8 (Method 2). All methods share the same network architecture as standard SAC. In adaptive meta warm-up, we perform the policy evaluation for 10 episodes among a random policy, the preceding fast policy, and the meta policy. Then we initialize the fast policy with the best-performing one. The collected data in evaluation is also stored in the fast learner's replay buffer $\mathcal{F}$ without incurring additional interaction costs with the environment. More experimental details of our `FAME` methods (5) and (6) are provided in Appendix G.1.

**Main Results.** As exhibited in Table 3, both `FAME-KL` and `FAME-WD` outperform most baselines significantly. `PackNet` achieves zero forgetting by storing (masks of) past model parameters and knowing the task identifiers and number in advance, which is less practical in real scenarios. By contrast, the superior forward transfer of FAME indicates that

Table 3: Main results on **Meta-World** on Average Performance (*Ave. Perf*), Forward Transfer (*FT*), and Forgetting averaged over 3 sequences. Results are presented as averages and standard errors across 10 seeds.

| Methods | Avg. Perf ↑ | FT ↑ | Forgetting ↓ |
|---|---|---|---|
| Reset | $0.093 \pm 0.017$ | $0.000 \pm 0.000$ | $0.710 \pm 0.030$ |
| Finetune | $0.037 \pm 0.011$ | $-0.265 \pm 0.028$ | $0.427 \pm 0.033$ |
| Average | $0.013 \pm 0.007$ | $-0.530 \pm 0.024$ | $0.070 \pm 0.022$ |
| PackNet | $0.491 \pm 0.025$ | $-0.194 \pm 0.018$ | $\mathbf{0.000 \pm 0.000}$ |
| FAME-KL | $0.733 \pm 0.026$ | $\mathbf{0.022 \pm 0.015}$ | $0.073 \pm 0.019$ |
| FAME-WD | $\mathbf{0.767 \pm 0.024}$ | $-0.003 \pm 0.014$ | $0.023 \pm 0.015$ |

the adaptive meta warm-up fosters the fast learner to adapt to a new environment by leveraging prior knowledge from the meta learner. Moreover, the highest average performance and almost minimal

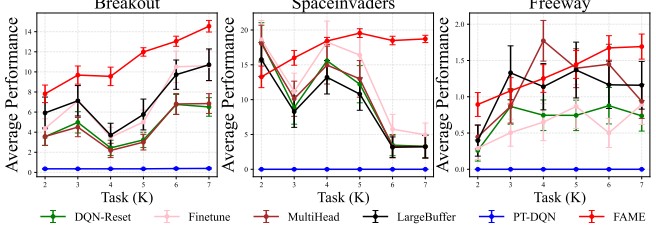
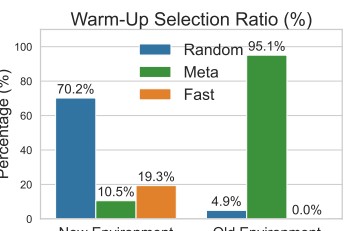

Figure 2: **(Left)** Average performance of the policy across each task across 10 sequences on **MinAtar**. Results are averaged over 3 seeds. The vertical lines at each point represent the standard errors. **(Right)** The selection ratio among three warm-up strategies when the arriving environment is previously encountered or novel.

forgetting of our `FAME` approaches highlight that the meta learner consolidates all past knowledge by conducting incremental updates in knowledge integration.

**Performance of Knowledge Transfer.** To comprehensively verify the knowledge transfer benefit due to the adaptive meta warm-up in `FAME`, we present the performance profile (Agarwal et al., 2021) that reflects the overall performance of the fast learner over the whole sequence of tasks. Figure 3 (left) suggests that both `FAME` methods outperform all baselines, substantiating that the meta learner effectively

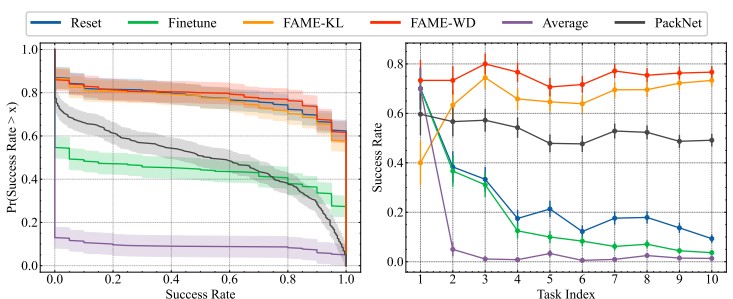

Figure 3: **(Left)** Performance profile of the fast learner across tasks, where the y-axis shows the proportion of tasks that achieve a success rate greater than or equal to the x-axis value. **(Right)** Average performance by evaluating the average success rates in past tasks across 10 seeds.

consolidates knowledge and enhances knowledge transfer. Learning curves and performance profiles for each sequence are also given in Appendix G.2.

**Performance of Knowledge Integration.** To reflect the tendency of the catastrophic forgetting of `FAME`, we also illustrate the average performance of the meta learner at the end of each task and then take the average over 3 sequences. As suggested in Figure 3 (right), `FAME-KL` and `FAME-MD` enjoy the highest average performance over time across all encountered tasks.

## 5 Discussions and Conclusion

In this paper, we contribute to the foundation of continual RL and develop a novel dual-learner algorithm to conduct the knowledge transfer and integration via the coupled update of fast and meta knowledge learners. Two ideas might be worth reemphasizing here. (1) Adaptively selecting practical prior knowledge (e.g., via the hypothesis test) is crucial to overcoming the negative transfer issue. (2) Deriving an incremental update rule based on existing multi-task learning objectives is necessary to connect continual and multi-task RL.

**Limitations and Future Work**. In this study, a meta learner is utilized to retain all knowledge. Alternatively, it is also possible to learn an effective latent representation that can not only distill all knowledge but also perform efficient reasoning to guide the adaptation to a new environment. Beyond the proposed adaptive meta warm-up, more techniques in knowledge transfer can be explored in the future, such as context embedding. Lastly, extending our algorithm to the full continual RL context without knowing the task boundary (possibly by developing online one-vs-all hypothesis test) or assuming the same state and action spaces (Hu et al., 2025) is also valuable for practitioners.

**Ethics Statement.** This study focuses on theoretical and algorithmic aspects of continual reinforcement learning, without involving human subjects, personal data, or sensitive applications. We acknowledge that continual learners deployed in real-world systems could, without careful control and monitoring, exhibit unexpected behaviors due to forgetting or transfer misalignment. However, our study remains purely theory- and algorithm-oriented, and we therefore do not foresee any direct ethical concerns arising from this research.

## Acknowledgements

Linglong Kong was partially supported by grants from the Canada CIFAR AI Chairs program, the Alberta Machine Intelligence Institute (AMII), the Natural Sciences and Engineering Council of Canada (NSERC), and the Canada Research Chair program from NSERC. Hongming Zhang was supported by the National Key Laboratory of Cognition and Decision Intelligence for Complex Systems, Institute of Automation, Chinese Academy of Sciences (Grant No. E5SPFZ0112). We also thank all the constructive suggestions and comments from the reviewers and area chairs.

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

# Appendix

## Table of Contents

# A    RELATED WORK

**Measure of MDP Distance and Catastrophic Forgetting.**    Although in a different perspective, Bisimulation metrics (Ferns & Precup, 2014) provides additional evidence to support the strategy that optimal value functions can be naturally used to measure the state similarity.

**Continual RL.**    Continual RL (Khetarpal et al., 2022; Abel et al., 2023) addresses the challenges in learning a sequence of decision-making tasks, particularly balancing the stability (i.e., mitigating catastrophic forgetting) and plasticity (i.e., rapid adaptation to a new environment). Below, we categorize the existing literature into two groups based on their focus and make a detailed comparison with our contribution in this study.

- **Catastrophic Forgetting**. **(1)** The replay-based approach is commonly applied to mitigate forgetting. A replay-based recurrent methodology was initially proposed for task-agnostic agents (Caccia et al., 2022). RECALL (Zhang et al., 2023) leverages adaptive normalization on approximate targets and policy distillation on old tasks to enhance generality and stability. Generative replay (Chen et al., 2024) was recently proposed using the diffusion model to memorize the high-return trajectory distribution of each encountered task. Sun et al. (2025) continually learns a generative model for experience replay without storing the past data within a model-based RL framework. Similarly, Liu et al. (2025) learns an online world model and acts by planning via model prediction control to construct a unified world dynamics to handle the catastrophic forgetting issue. **(2)** The second branch is regularization-based from distinct perspectives. Earlier, the behavioral cloning was investigated across historical policies in (Wolczyk et al., 2022). Inspired by complex synapses, (Kaplanis et al., 2018; 2019) developed the policy consolidation strategy by simultaneously remembering the agent's policy at a range of timescales and regularizing the current policy by its own historical experience. Sparse prompting (Yang et al., 2023) imposes a regularization via dictionary learning to produce sparse masks as prompts, extracting a sub-network for each task from a meta-policy network. **(3)** Model expansion also serves as a promising direction to investigate (Mallya & Lazebnik, 2018; Malagon et al., 2024; Gaya et al., 2023). Gaya et al. (2023) builds the subspace of policies to consider the scalable continual RL, while pointing out the trade-off between the agent's size and the performance of continual learning. Malagon et al. (2024) uses a growing policy neural network and applies the attention mechanism to integrate the knowledge from the previous policies and the current state to "self-compose" an internal policy. Despite the effectiveness of model expansion-based approaches, the primary concerns lie in their high memory and inference costs due to the leverage of previous policies with specific aggregated strategies. For instance, the attention module is adopted in (Malagon et al., 2024), where the number of parameters grows *linearly* and the theoretical computational cost is *quadratic* with respect to the number of tasks. **Comparison:** Our FAME approach with a dual-learner strategy can be viewed as lying at the intersection of the three predominant paradigms. In the knowledge integration, minimizing catastrophic forgetting requires an experience replay from the meta buffer. In the knowledge transfer, a behavior cloning regularization is possibly adopted to guide the exploration if the meta learner is chosen as the best in the adaptive meta warm-up framework. Although employing a dual-learner strategy, FAME retains the fixed model sizes of both fast and meta learners to avoid the stability issue, which is distinct from other model expansion approaches.

- **Knowledge Transfer and Loss of Plasticity**. The effectiveness of knowledge forward transfer determines the loss of plasticity, a reduced capability to rapidly adapt to a new environment (Abel et al., 2018; Dohare et al., 2024; Wolczyk et al., 2022; Tao et al., 2021). Within the continual RL literature, a large number of studies mainly focus on the knowledge transfer and plasticity capability through distinct perspectives. Xie & Finn (2022) improves the forward transfer and mitigates the loss of plasticity by selectively identifying the most relevant samples for the new task using learnable importance weights. The value function decomposing approach (Anand & Precup, 2023) is proposed to perform an interplay between fast and slow learning at various levels for value-based continual RL with a discrete action space to address the loss of plasticity. Through the lens of optimization, Parseval network (Chung et al., 2024) imposes orthogonality constraints to mitigate interference, while Muppidi et al. (2024) employs parameter-free online convex optimization to retain plasticity. A recent work by (Tang et al., 2025) establishes the connection between plasticity and the *churn* via the Neural Tangent Kernel (NTK) matrix. *Negative transfer* issue (Ahn et al., 2025) was revealed in the adaptation to a new task due to the task dissimilarity, amplifying the loss of plasticity. As such, Reset and Distill (R&D) (Ahn et al., 2025) explicitly

resets the policy in each new environment to avoid the negative transfer. **Comparison:** In terms of the knowledge transfer, it is not necessary that the reset is always the best choice, especially when the task similarity often exists. By leveraging the one-vs-all hypothesis test, we propose the adaptive meta warm-up approach that selectively discriminates the most effective weight initialization and warm-up strategy to facilitate the knowledge transfer and reduce the loss of plasticity.

**Transfer, Multi-task, and Meta RL.** The knowledge transfer phase in our approach is closely linked with transfer RL, where the accrued knowledge can be transferred through representation (Rusu et al., 2016; Devin et al., 2017), learned models (Finn & Levine, 2017; Eysenbach et al., 2021), and network weights (Fernández & Veloso, 2006) or experience (Andrychowicz et al., 2017; Tirinzoni et al., 2018; Xie & Finn, 2022). Although the set of tasks should be learned simultaneously, multi-task RL also integrates the component of knowledge transfer, such as (Caruana, 1997; Sodhani et al., 2021; Teh et al., 2017; Yang et al., 2023; Rusu et al., 2015; Hausman et al., 2018). As such, transfer RL has been an underpinning building block for both multi-task and continual RL, e.g., the explicit knowledge transfer in our dual-learner algorithm. As opposed to multi-task RL, the incremental or sequential learning nature of continual RL makes it more challenging to minimize catastrophic forgetting. However, our study shows that minimizing catastrophic forgetting can be, in principle, equivalent to the objectives in multi-task learning. Meta RL (Finn et al., 2017) can be seen as an extension or generalization of multi-task RL, with explicit mechanisms for fast adaptation and few-shot learning, and is also closely linked with continual RL. Continual Meta-Policy Search (CoMPS) (Berseth et al., 2022) probes the setting of meta-training in an incremental fashion, extending meta-RL to a continual learning scenario. **Comparison:** The fast and meta learners adopted in our FAME approach intermingle the key techniques of transfer, multi-task, and meta RL, illuminating their deep connections within our algorithmic framework. The interplay of knowledge transfer and knowledge integration via the coupled updates between fast and meta learners simultaneously tackles the involved challenges, exhibiting promising solutions to address continual RL.

## B   PRELIMINARIES: DEFINITION OF DISTRIBUTION DRIFT AND CATASTROPHIC FORGETTING IN DEEP LEARNING

We first introduce the concept of *drift* in the process of learning a parameterized function $f$ from the source data distribution $\tau_S$ with the dataset $\mathcal{D}_{\tau_S}$ to the target data distribution $\tau_T$ with the dataset $\mathcal{D}_{\tau_T}$. After learning $f$ on the source dataset $\mathcal{D}_{\tau_S}$, we obtain the estimated function $\widehat{f}_{\tau_S}$. Then we apply the same model architecture $f$ on the target dataset $\mathcal{D}_{\tau_T}$ with any learning algorithms, and finally we evaluate the drift of the attained $\widehat{f}_{\tau_T}$ via $\delta^{\tau_S \to \tau_T}$ defined as (Doan et al., 2021):

$$\delta^{\tau_S \to \tau_T}\left(X^{\tau_S}\right) = \left(\widehat{f}_{\tau_T}(x) - \widehat{f}_{\tau_S}(x)\right)_{(x,y)\in\mathcal{D}_{\tau_S}} \tag{10}$$

Based on the definition of *drift*, we define the *vanilla catastrophic forgetting* $\Delta^{\tau_S \to \tau_T}$ as

$$\Delta^{\tau_S \to \tau_T}\left(X^{\tau_S}\right) = \left\|\delta^{\tau_S \to \tau_T}\left(X^{\tau_S}\right)\right\|_2^2 = \sum_{(x,y)\in\mathcal{D}_{\tau_S}} \left(\widehat{f}_{\tau_T}(x) - \widehat{f}_{\tau_S}(x)\right)^2, \tag{11}$$

where the catastrophic forgetting can be further simplified as $\Delta^{\tau_S \to \tau_T} = \left\|\phi\left(X^{\tau_S}\right)\left(\omega_{\tau_T}^* - \omega_{\tau_S}^*\right)\right\|_2^2$ in the Neural Tangent Kernel (NTK) regime (Doan et al., 2021; Jacot et al., 2018), allowing the proposal of new continual learning approaches. In deep learning, minimizing the catastrophic forgetting $\Delta^{\tau_S \to \tau_T}$ is equivalent to minimizing a weighted drift in terms of the prediction function $\widehat{f}$ with the weights determined by the dataset. The preliminary results in deep learning largely inspire us to explore the foundations in continual RL.

## C   THEORETICAL RESULTS

### C.1   INCREMENTAL Q-VALUE-BASED META LEARNER UPDATE IN PROPOSITION 3

In Proposition 3, we provide an efficient incremental update rule of the meta learner to minimize the principled Q-value-based catastrophic forgetting that we define in Definition 2.

**Proposition 3** (Incremental Q-Value-based Meta Learner Update). *Let $d_Q$ be $\ell_2$ loss in Eq. 1 in Definition 2. Minimizing Q-value-based catastrophic forgetting in Eq. 3 is equivalent to:*

$$Q_k^M = \arg\min_{\widetilde{Q}_k^M} \sum_{i=1}^{k-1} \mathbb{E}_{w_i^Q}\left[\left(Q_{k-1}^M - \widetilde{Q}_k^M\right)^2\right] + \mathbb{E}_{w_k^Q}\left[\left(Q_k - \widetilde{Q}_k^M\right)^2\right]. \tag{12}$$

*Proof.* **Step 1: Optimality Condition.** Recap $w_i^Q(s,a) = \mu_i^{Q_i}(s)\pi^{Q_i}(a|s)$. We aim to minimize the Q-value-based catastrophic forgetting defined in Eq. 3:

$$Q_k^M = \arg\min_{\widetilde{Q}_k^M} \sum_{i=1}^{k} \sum_{s,a} w_i^Q(s,a)\left(Q_i(s,a) - \widetilde{Q}_k^M(s,a)\right)^2.$$

For each $s$ and $a$, by taking the derivative of the objective in Eq. 3 regarding $\widetilde{Q}_k^M$, the first-order optimality condition is

$$\sum_{i=1}^{k} w_i^Q(s,a)\left(Q_k^M(s,a) - Q_i(s,a)\right) = 0. \tag{13}$$

By rewriting the two optimality conditions regarding $Q_k^M$ and $Q_{k-1}^M$, we can attain that

$$Q_k^M(s,a) = \frac{\sum_{i=1}^{k} w_i^Q(s,a)Q_i(s,a)}{\sum_{j=1}^{k} w_j^Q(s,a)}, \quad Q_{k-1}^M(s,a) = \frac{\sum_{i=1}^{k-1} w_i^Q(s,a)Q_i(s,a)}{\sum_{j=1}^{k-1} w_j^Q(s,a)}.$$

**Step 2: Incremental Update Rule.** For brevity, we employ the expectation operation $\mathbb{E}_w$. Based on the two optimality conditions above, we can derive the following incremental update rule:

$$
\begin{aligned}
Q_k^M &= \arg\min_{\widetilde{Q}_k^M} \sum_{i=1}^{k} \mathbb{E}_{w_i^Q}\left[\left(Q_i - \widetilde{Q}_k^M\right)^2\right] \\
&= \arg\min_{\widetilde{Q}_k^M} \underbrace{\sum_{i=1}^{k-1} \mathbb{E}_{w_i^Q}\left[\left(Q_i - Q_{k-1}^M + Q_{k-1}^M - \widetilde{Q}_k^M\right)^2\right]}_{\text{①}} + \mathbb{E}_{w_k^Q}\left[\left(Q_k - \widetilde{Q}_k^M\right)^2\right]. \tag{14}
\end{aligned}
$$

$$
\begin{aligned}
\text{①} &= \sum_{i=1}^{k-1} \mathbb{E}_{w_i^Q}\left[\left(Q_i - Q_{k-1}^M\right)^2 + \left(Q_{k-1}^M - \widetilde{Q}_k^M\right)^2 + 2\left(Q_i - Q_{k-1}^M\right)\left(Q_{k-1}^M - \widetilde{Q}_k^M\right)\right] \\
&= \sum_{i=1}^{k-1} \mathbb{E}_{w_i^Q}\left[\left(Q_i - Q_{k-1}^M\right)^2 + \left(Q_{k-1}^M - \widetilde{Q}_k^M\right)^2\right] + \\
&\quad 2\sum_{s,a}\left(Q_{k-1}^M(s,a) - \widetilde{Q}_k^M(s,a)\right)\left(\sum_{i=1}^{k-1} w_i^Q(s,a)\left(Q_i(s,a) - Q_{k-1}^M(s,a)\right)\right) \\
&\stackrel{(a)}{=} \sum_{i=1}^{k-1} \mathbb{E}_{w_i^Q}\left[\left(Q_i - Q_{k-1}^M\right)^2 + \left(Q_{k-1}^M - \widetilde{Q}_k^M\right)^2\right] \\
&= C + \sum_{i=1}^{k-1} \mathbb{E}_{w_i^Q}\left[\left(Q_{k-1}^M - \widetilde{Q}_k^M\right)^2\right],
\end{aligned}
$$

where $(a)$ holds as $\sum_{i=1}^{k-1} w_i^Q(s,a)\left(Q_i(s,a) - Q_{k-1}^M(s,a)\right) = 0$ is the optimality condition for $Q_{k-1}^M$ in Eq. 13 of Step 1. $C = \sum_{i=1}^{k-1} \mathbb{E}_{w_i^Q}\left[\left(Q_i - Q_{k-1}^M\right)^2\right]$ is a constant in terms of $\widetilde{Q}_k^M$. Putting all together into Eq. 14, we have

$$Q_k^M = \arg\min_{\widetilde{Q}_k^M} \sum_{i=1}^{k-1} \mathbb{E}_{w_i^Q}\left[\left(Q_{k-1}^M - \widetilde{Q}_k^M\right)^2\right] + \mathbb{E}_{w_k^Q}\left[\left(Q_k - \widetilde{Q}_k^M\right)^2\right].$$

$\square$

## C.2 PROOF OF PROPOSITION 1

**Proposition** 1 [Incremental Softmax Q-Value-based Meta Learner Update] Denote $\pi_k^M(a|s) = \exp\left(Q_k^M(a|s)/\tau\right)/\sum_{a'}\exp\left(Q_k^M(a'|s)/\tau\right)$. After a softmax policy transformation, the Q-value-based meta learner incremental update is rewritten as

$$Q_k^M = \arg\min_{\widetilde{Q}_k^M} \sum_{i=1}^{k-1} \mathbb{E}_{w_i^Q}\left[\log\frac{\pi_{k-1}^M}{\widetilde{\pi}_k^M}\right] + \mathbb{E}_{w_k^Q}\left[\log\frac{\pi^{Q_k}}{\widetilde{\pi}_k^M}\right] = \arg\max_{\widetilde{Q}_k^M} \sum_{i=1}^{k} \mathbb{E}_{w_i^Q}\left[\log\widetilde{\pi}_k^M\right],$$

*Proof.* We rely on the softmax transformation to transfer a meta Q function to a meta policy. As such, the policy-based catastrophic forgetting in Eq. 4, when adapted from value-based continual RL and equipped with KL divergence as $d_\pi$, can be expressed as

$$Q_k^M = \arg\min_{\widetilde{Q}_k^M} \sum_{i=1}^{k} \sum_{s} \mu_i^{Q_i}(s)\left[\text{KL}\left(\pi^{Q_i}(\cdot|s)||\widetilde{\pi}_k^M(\cdot|s)\right)\right]. \tag{15}$$

where $\widetilde{\pi}_k^M(a|s) = \exp\left(\widetilde{Q}_k^M(a|s)/\tau\right)/\sum_{a'}\exp\left(\widetilde{Q}_k^M(a'|s)/\tau\right)$. By the definition of the KL divergence, we can rewrite the objective function in Eq. 15 as an incremental update rule:

$$
\begin{aligned}
Q_k^M &= \arg\min_{\widetilde{Q}_k^M} \sum_{i=1}^{k} \sum_{s,a} \mu_i^{Q_i}(s)\pi^{Q_i}(a|s)\log\frac{\pi^{Q_i}(a|s)}{\widetilde{\pi}_k^M(a|s)} \\
&= \arg\min_{\widetilde{Q}_k^M} \sum_{i=1}^{k} \mathbb{E}_{w_i^Q}\left[\log\frac{\pi^{Q_i}}{\widetilde{\pi}_k^M}\right] \\
&= \arg\min_{\widetilde{Q}_k^M} \sum_{i=1}^{k-1} \mathbb{E}_{w_i^Q}\left[\log\left(\frac{\pi^{Q_i}}{\widetilde{\pi}_k^M}\frac{\pi_{k-1}^M}{\pi_{k-1}^M}\right)\right] + \mathbb{E}_{w_k^Q}\left[\log\frac{\pi^{Q_k}}{\widetilde{\pi}_k^M}\right] \\
&= \arg\min_{\widetilde{Q}_k^M}\left\{\sum_{i=1}^{k-1} \mathbb{E}_{w_i^Q}\left[\log\frac{\pi_{k-1}^M}{\widetilde{\pi}_k^M}\right] + \mathbb{E}_{w_k^Q}\left[\log\frac{\pi^{Q_k}}{\widetilde{\pi}_k^M}\right]\right\} + C \\
&= \arg\min_{\widetilde{Q}_k^M}\left\{\sum_{i=1}^{k-1} \mathbb{E}_{w_i^Q}\left[\log\frac{\pi_{k-1}^M}{\widetilde{\pi}_k^M}\right] + \mathbb{E}_{w_k^Q}\left[\log\frac{\pi^{Q_k}}{\widetilde{\pi}_k^M}\right]\right\} \\
&= \arg\min_{\widetilde{Q}_k^M}\left\{\sum_{i=1}^{k-1} \mathbb{E}_{w_i^Q}\left[\log\frac{1}{\widetilde{\pi}_k^M}\right] + \mathbb{E}_{w_k^Q}\left[\log\frac{1}{\widetilde{\pi}_k^M}\right]\right\} \\
&= \arg\max_{\widetilde{Q}_k^M} \sum_{i=1}^{k} \mathbb{E}_{w_i^Q}\left[\log\widetilde{\pi}_k^M\right],
\end{aligned}
\tag{16}
$$
$$\tag{17}$$

where $C = \sum_{i=1}^{k-1} \mathbb{E}_{w_i^Q}\left[\log\frac{\pi^{Q_i}}{\pi_{k-1}^M}\right]$ is a constant and is independent of $\widetilde{Q}_k^M$. Although it may be trivial to keep the form of Eq. 16, it emphasizes an incremental update rule of $Q_k^M$ based on $Q_{k-1}^M$ ($\pi_{k-1}^M$) and $Q_k$ ($\pi^{Q_k}$). Eventually, this minimization leads to an Maximum Likelihood estimation regarding the meta learner $\widetilde{Q}_k^M$ in Eq. 17, on a mixture of state-action distribution of all encountered environments up to $k$.

□

## C.3 PROOF OF PROPOSITION 2

**Proposition 2** [Incremental Policy-based Meta Learner Update under Wasserstein Distance] Consider $d_\pi$ to be the squared 2-Wasserstein distance in Eq. 2 of Definition 2 and the policy is represented as an independent (multivariate) Gaussian distribution over the action $a$. Minimizing policy-

based catastrophic forgetting in Eq. 4 is equivalent to:

$$\pi_{\mathrm{M}}^k = \arg\min_{\widetilde{\pi}_k^M} \left\{ \sum_{i=1}^{k-1}\sum_s \mu_i^{\pi_i}(s)W_2^2\left(\widetilde{\pi}_k^M(\cdot|s),\pi_{k-1}^M(\cdot|s)\right) + \sum_s \mu_k^{\pi_k}(s)W_2^2\left(\widetilde{\pi}_k^M(\cdot|s),\pi_k(\cdot|s)\right) \right\}.$$

*Proof.* Recap the objective of the policy-based catastrophic forgetting based on Eq. 4 under squared 2-Wasserstein distance:

$$\pi_k^M = \arg\min_{\widetilde{\pi}_k^M} \sum_{i=1}^{k}\sum_s \mu_i^{\pi_i}(s)W_2^2\left(\pi_i(\cdot|s),\widetilde{\pi}_k^M(\cdot|s)\right).$$

where the squared 2-Wasserstain distance between two Gaussian distributions $p$ and $q$ has a closed-form solution:

$$W_2^2(p,q) = \|\nu_p - \nu_q\|_2^2 + \mathrm{tr}\left(\Sigma_p + \Sigma_q - 2\left(\Sigma_q^{1/2}\Sigma_p\Sigma_q^{1/2}\right)^{1/2}\right), \tag{18}$$

with the two Gaussian distributions denoted by $\mathcal{N}(\nu_p,\Sigma_p)$ and $\mathcal{N}(\nu_q,\Sigma_q)$. In particular, when the policy is represented as an independent (multivariate) Gaussian distribution across the action $a$, it implies that $\Sigma_p$ and $\Sigma_q$ are diagonal (i.e., variables are independent), then the squared 2-Wasserstain distance in Eq. 18 can be further simplified as

$$W_2^2(p,q) = \|\nu_p - \nu_q\|_2^2 + \|\sigma_p - \sigma_q\|_2^2, \tag{19}$$

where $\sigma_p$ and $\sigma_q$ are the diagonal vector of $\Sigma_p$ and $\Sigma_q$, respectively. Then, the objective of the policy-based catastrophic forgetting based on Eq. 4 can be simplified as

$$\pi_k^M = \arg\min_{\widetilde{\nu}_k^M,\widetilde{\sigma}_k^M} \sum_{i=1}^{k}\sum_s \mu_i^{\pi_i}(s)\left(\|\nu_i(s) - \widetilde{\nu}_k^M(s)\|_2^2 + \|\sigma_i(s) - \widetilde{\sigma}_k^M(s)\|_2^2\right), \tag{20}$$

where $\pi_i(\cdot|s)$ is represented as a (multivariate) Gaussian distribution $\mathcal{N}(\nu_i(s),\sigma_i^2(s))$, where $\nu_i(s)$ and $\sigma_i^2(s)$ are the mean (vector) and (the diagonal vector of) the variance. Similarly, $\pi_M^k$ is represented as a (multivariate) Gaussian distribution $\mathcal{N}(\nu_k^M(s),(\sigma_k^M(s))^2)$.

**Step 1: Optimality Condition.** For each $s$, we take the derivative of Eq. 20 in terms of $\widetilde{\nu}_k^M$ and $\widetilde{\sigma}_k^M$, respectively. Consequently, it arrives at the following optimality condition:

$$\sum_{i=1}^{k}\mu_i^{\pi_i}(s)\left(\nu_i(s) - \nu_k^M(s)\right) = 0 \tag{21}$$

$$\sum_{i=1}^{k}\mu_i^{\pi_i}(s)\left(\sigma_i(s) - \sigma_k^M(s)\right) = 0. \tag{22}$$

**Step 2: Incremental Update.** We first rewrite Eq. 20 as

$$\pi_k^M = \arg\min_{\widetilde{\nu}_k^M,\widetilde{\sigma}_k^M} \underbrace{\sum_{i=1}^{k-1}\sum_s \mu_i^{\pi_i}(s)\|\nu_i(s) - \widetilde{\nu}_k^M(s)\|_2^2 + \sum_s \mu_k^{\pi_k}(s)\|\nu_k(s) - \widetilde{\nu}_k^M(s)\|_2^2}_{\textcircled{1}}$$

$$+ \underbrace{\sum_{i=1}^{k-1}\sum_s \mu_i^{\pi_i}(s)\|\sigma_i(s) - \widetilde{\sigma}_k^M(s)\|_2^2 + \sum_s \mu_k^{\pi_k}(s)\|\sigma_k(s) - \widetilde{\sigma}_k^M(s)\|_2^2}_{\textcircled{2}}.$$

$$\textcircled{1} = \sum_{i=1}^{k-1} \sum_s \mu_i^{\pi_i}(s) \| \nu_i(s) - \nu_{k-1}^M(s) + \nu_{k-1}^M(s) - \widetilde{\nu}_k^M(s) \|_2^2$$

$$= \sum_{i=1}^{k-1} \sum_s \mu_i^{\pi_i}(s) \left( \| \nu_i(s) - \nu_{k-1}^M(s) \|_2^2 + \| \nu_{k-1}^M(s) - \widetilde{\nu}_k^M(s) \|_2^2 \right) +$$

$$2 \sum_{i=1}^{k-1} \sum_s \mu_i^{\pi_i}(s) \langle \nu_i(s) - \nu_{k-1}^M(s), \nu_{k-1}^M(s) - \widetilde{\nu}_k^M(s) \rangle$$

$$= \sum_{i=1}^{k-1} \sum_s \mu_i^{\pi_i}(s) \left( \| \nu_i(s) - \nu_{k-1}^M(s) \|_2^2 + \| \nu_{k-1}^M(s) - \widetilde{\nu}_k^M(s) \|_2^2 \right) +$$

$$2 \sum_s \langle \sum_{i=1}^{k-1} \mu_i^{\pi_i}(s) \left( \nu_i(s) - \nu_{k-1}^M(s) \right), \nu_{k-1}^M(s) - \widetilde{\nu}_k^M(s) \rangle$$

$$\overset{(a)}{=} \sum_{i=1}^{k-1} \sum_s \mu_i^{\pi_i}(s) \left( \| \nu_i(s) - \nu_{k-1}^M(s) \|_2^2 + \| \nu_{k-1}^M(s) - \widetilde{\nu}_k^M(s) \|_2^2 \right),$$

where $(a)$ holds due to the optimality condition $\sum_{i=1}^{k-1} \mu_i^{\pi_i}(s) \left( \nu_i(s) - \nu_{k-1}^M(s) \right) = 0$ we derived in Eq. 21 of Step 1. Similarly, we can show this simplification regarding the variance:

$$\textcircled{2} = \sum_{i=1}^{k-1} \sum_s \sigma_i^{\pi_i}(s) \| \sigma_i(s) - \sigma_{k-1}^M(s) + \sigma_{k-1}^M(s) - \widetilde{\sigma}_k^M(s) \|_2^2$$

$$= \sum_{i=1}^{k-1} \sum_s \mu_i^{\pi_i}(s) \left( \| \sigma_i(s) - \sigma_{k-1}^M(s) \|_2^2 + \| \sigma_{k-1}^M(s) - \widetilde{\sigma}_k^M(s) \|_2^2 \right) +$$

$$2 \sum_{i=1}^{k-1} \sum_s \mu_i^{\pi_i}(s) \langle \sigma_i(s) - \sigma_{k-1}^M(s), \sigma_{k-1}^M(s) - \widetilde{\sigma}_k^M(s) \rangle$$

$$= \sum_{i=1}^{k-1} \sum_s \mu_i^{\pi_i}(s) \left( \| \sigma_i(s) - \sigma_{k-1}^M(s) \|_2^2 + \| \sigma_{k-1}^M(s) - \widetilde{\sigma}_k^M(s) \|_2^2 \right) +$$

$$2 \sum_s \langle \sum_{i=1}^{k-1} \mu_i^{\pi_i}(s) \left( \sigma_i(s) - \sigma_{k-1}^M(s) \right), \sigma_{k-1}^M(s) - \widetilde{\sigma}_k^M(s) \rangle$$

$$\overset{(b)}{=} \sum_{i=1}^{k-1} \sum_s \mu_i^{\pi_i}(s) \left( \| \sigma_i(s) - \sigma_{k-1}^M(s) \|_2^2 + \| \sigma_{k-1}^M(s) - \widetilde{\sigma}_k^M(s) \|_2^2 \right),$$

where $(b)$ holds due to the optimality condition $\sum_{i=1}^{k-1} \mu_i^{\pi_i}(s) \left( \sigma_i(s) - \sigma_{k-1}^M(s) \right) = 0$ we derived in Eq. 22 of Step 1. Putting all together, we have

$$\pi_k^M = \underset{\widetilde{\nu}_k^M, \widetilde{\sigma}_k^M}{\arg\min} \sum_{i=1}^{k-1} \sum_s \mu_i^{\pi_i}(s) \left( \| \nu_i(s) - \nu_{k-1}^M(s) \|_2^2 + \| \nu_{k-1}^M(s) - \widetilde{\nu}_k^M(s) \|_2^2 \right) + \sum_s \mu_k^{\pi_k}(s) \| \nu_k(s) - \widetilde{\nu}_k^M(s) \|_2^2$$

$$+ \sum_{i=1}^{k-1} \sum_s \mu_i^{\pi_i}(s) \left( \| \sigma_i(s) - \sigma_{k-1}^M(s) \|_2^2 + \| \sigma_{k-1}^M(s) - \widetilde{\sigma}_k^M(s) \|_2^2 \right) + \sum_s \mu_k^{\pi_k}(s) \| \sigma_k(s) - \widetilde{\sigma}_k^M(s) \|_2^2$$

By removing the constant terms independent of $\widetilde{\nu}_k^M$ and $\widetilde{\sigma}_k^M$, we further have

$$
= \underset{\widetilde{\nu}_k^M, \widetilde{\sigma}_k^M}{\arg\min} \sum_{i=1}^{k-1} \sum_s \mu_i^{\pi_i}(s) \|\nu_{k-1}^M(s) - \widetilde{\nu}_k^M(s)\|_2^2 + \sum_s \mu_k^{\pi_k}(s) \|\nu_k(s) - \widetilde{\nu}_k^M(s)\|_2^2
$$

$$
+ \sum_{i=1}^{k-1} \sum_s \mu_i^{\pi_i}(s) \|\sigma_{k-1}^M(s) - \widetilde{\sigma}_k^M(s)\|_2^2 + \sum_s \mu_k^{\pi_k}(s) \|\sigma_k(s) - \widetilde{\sigma}_k^M(s)\|_2^2
$$

$$
= \underset{\widetilde{\pi}_k^M}{\arg\min} \left\{ \sum_{i=1}^{k-1} \sum_s \mu_i^{\pi_i}(s) W_2^2 \left( \widetilde{\pi}_k^M(\cdot|s), \pi_{k-1}^M(\cdot|s) \right) + \sum_s \mu_k^{\pi_k}(s) W_2^2 \left( \widetilde{\pi}_k^M(\cdot|s), \pi_k(\cdot|s) \right) \right\}.
$$

This leads to the incremental policy-based meta learner update under the squared 2-Wasserstein distance.

$\square$

## D  POLICY-BASED FAME ALGORITHM

**Algorithm Description.** We first denote the fast buffer as $\mathcal{F}$ and $\pi^0$ as the initialized policy. As suggested in Algorithm 1, when the $k$-th environment arrives, we initialize the fast learner $\pi_k$ via the adaptive meta warm-up among the preceding meta learner $\pi_{k-1}^M$, the preceding fast learner $\pi_{k-1}$ and a random learner $\pi^0$ (reset strategy) within $L$ steps. The adaptive meta warm-up makes full use of previous information to perform an effective knowledge transfer. Once the $k$-th task ends, the knowledge integration phase starts, when the meta learner $\pi_k^M$ is updated via Eq. 7 (FAME-KL) or via Eq. 8 (FAME-MD) on the data collected in the meta buffer $\mathcal{M}$. The meta learner incrementally incorporates the knowledge from $\pi_k$ into $\pi_{k-1}^M$, leading to an updated meta learner $\pi_k^M$.

---

**Algorithm 2** Policy-based FAME Update in the $k$-th Environment

---

1: **Initialize**: Fast Buffer $\mathcal{F}$, Meta Buffer $\mathcal{M}$, $\pi_{k-1}^M$, $\pi_{k-1}$, $\pi^0$, Warm-Up Step $L$, Estimation Step $N$.
2: # Knowledge Transfer: Adaptive Meta Warm-Up
3: Initialize $\pi_k$ in $\{\pi_{k-1}, \pi_k^M, \pi^0\}$ via Eq. 6 after $L$ steps
4: **for** $t = L$ to $T$ **do**
5:      Observe $S_t$, take action $A_t$, receive $R_t$, observe $S_{t+1}$
6:      Store $(S_t, A_t, R_t, S_{t+1})$ in $\mathcal{F}$
7:      Update $\pi_k$
8:      **if** $t > T - N$ **then**
9:          Method 1 (FAME-KL): Store $(S_t, A_t)$ in $\mathcal{M}$ # To Estimate $w_k$
10:          Method 2 (FAME-WD): Store $S_t$ in $\mathcal{M}$ # To Estimate $\mu_k^{\pi_k}$
11:      **end if**
12: **end for**
13: Reset $\mathcal{F}$
14: # Knowledge Integration: Minimize Catastrophic Forgetting
15: Method 1 (FAME-KL): Update $\pi_k^M$ via Eq. 7 on state-action pairs in $\mathcal{M}$
16: Method 2 (FAME-WD): Update $\pi_k^M$ via Eq. 8 on states in $\mathcal{M}$

---

**Computational Cost, Performance Variability, and Practical Guidance.** The increased computational overhead of FAME-WD is negligible compared to FAME-KL when using standard Gaussian policy parameterizations, which is common in RL in a continuous action space. As exhibited in Eq. 19, the simplification of Wasserstein distance leads to the same output of the policy network, ensuring a comparable computational cost to the KL divergence. We also remark that although the Wasserstein distance better captures the data geometry, the Gaussian policy may restrict the Wasserstein distance's advantage. Therefore, FAME-WD does not necessarily outperform FAME-KL. In complex environments, the task involves high distributional shift and the change of stochasticity, where capturing distribution geometry becomes critical. We believe that is when FAME-WD becomes potentially superior to FAME-KL.

# E    DISCUSSION OF THE FAME FRAMEWORK

**Scalability of Storing Past Trajectories in the Meta Buffer.**    As described in the value-based FAME algorithm, at the end of training in each environment, we store a subset of state–action pairs to estimate the weight $w_k$. Importantly, the stored pairs constitute only a small fraction of the total training data for each task, approximately 1–2% in our experiments. Consequently, our FAME approach incurs negligible scalability overhead on the benchmarks considered. For a fair comparison, we maintain the meta buffer at the same size as the replay buffer used in standard RL, and our FAME approaches still exhibit superior performance across all considered benchmarks. Still, as shown in Table 6 in Appendix F.2, enlarging the meta buffer by including more past trajectories generally leads to improved performance. In addition, as the meta learning update is in a supervised learning manner, which is much cheaper than the online RL training, even across a larger training set in the meta buffer. Overall, FAME achieves strong performance with minimal scalability concerns, highlighting its potential to effectively address key challenges in continual reinforcement learning.

**Principled Strategy for Determining $L$ for the Policy Evaluation.**    Regarding the choice of the warm-up step $L$, the optimal $L$ is indeed task-dependent. As increasing $L$ will reduce the number of available samples for the online learning, given a fixed total number of interaction steps, the optimal $L$ needs to strike a balance between selecting the most effective prior knowledge and preserving sufficient data for online adaptation. However, in the Meta-World experiments, the warm-up evaluation was simply fixed at 10 episodes because this choice already provides strong performance and significantly outperforms other baselines. In value-based continual RL, a longer warm-up duration $L$ in the behavior cloning regularization does not necessarily improve overall learning. When the meta learner is selected to warm up the fast learner via the one-vs-all hypothesis test, the conclusion is that the prior knowledge from the meta learner provides a better initialization than either a random learner or the preceding fast learner, and is able to accelerate the training process especially in the early stages of training. However, as learning progresses, the fast learner gradually adapts to the new environment and can surpass the meta learner's performance. In such cases, a prolonged behavior cloning regularization tends to over-constrain the fast learner, thereby hindering adaptation and leading to the degradation in final performance (e.g., the non-monotonic effect in Table 5).

**Principled Strategy for Determining $N$.**    Regarding the number of stored trajectories $N$, its selection primarily depends on computational and memory constraints (i.e., the meta buffer size). As shown in Table 6 of Appendix F.2, increasing $N$ generally improves performance but comes with higher costs in both memory (increasing the meta buffer size) and computation (meta learner update on a larger trajectory dataset) in the knowledge integration phase. However, as discussed previously, FAME achieves strong performance with minimal scalability concerns and we can increase $N$ in practice to pursue superior performance as long as the memory budget is allowed.

**Space and Computational Complexity of FAME.** (1) In terms of space complexity, the dual learner system of FAME requires an additional memory copy as the fast learner (the normal learner in baselines such as `Reset` and `Finetune`), and an additional meta replay buffer with the same size as the buffer of the fast learner, which is scalable as the total sample memory is fixed and independent of the number of future tasks. (2) In terms of the computational cost, we employ the same number of agent's interaction steps with the environment for a fair comparison. In other words, the policy evaluation occurs at the cost of reducing the policy optimization update steps in total. The main additional computation cost is the updating of the meta-learner, which is efficiently conducted in a supervised learning way. For instance, we found it only takes around a couple of minutes across 200 epochs to update the meta-learners in MinAtar after one task finishes. This additional computation overhead is negligible to the overall computation cost in the online training of RL algorithms.

# F  EXPERIMENTS: VALUE-BASED CONTINUAL RL WITH DISCRETE ACTION SPACE

## F.1  DETAILS OF EXPERIMENTAL SETUP AND COMPARISON METHODS

**Metric Calculation.** As average performance is the main metric in continual RL, we report the results for each environment. For the evaluation of forgetting, we first calculate the metric scores for each environment and then normalize them by their standard deviation across all methods in each environment. This standard normalization mitigates the influence of different reward scales of each game and allows us to average them across games to report a more comprehensive forgetting score.

**Hyperparameters: MinAtar.** Our implementation adapts from the released code in (Anand & Precup, 2023). For our FAME approach, after doing the line search of the hyperparameter in Section F.2, we choose the estimation step $N = 12000$, i.e., the number of data to be stored in the meta buffer $\mathcal{M}$ with size 100000 in each task, the policy evaluation step $n = 600$ for one-vs-all hypothesis test, and the warm-up step $L = 50000$ (10% of the training steps in each task). *Note that we use $L$ to represent the warm-up steps in the learning with the behavior cloning regularization, while introducing another notation $n$ to denote the policy evaluation steps for adaptive warm-up.* The ratio of stored data across the whole training steps in each task is $12000/500\text{k} = 2.4\%$. In the knowledge integration phase, we train the meta learner across 200 epochs from a $1 \times 10^{-3}$ learning rate with a decaying strategy. The learning rate for the fast learner is kept as $1 \times 10^{-5}$, the same as the other variants of DQN baselines. Every time a new environment arrives, we clear the fast buffer $\mathcal{F}$ and reinitialize the parameters of all involved learners, except for DQN-Finetune. For FAME, after the adaptive meta warm-up, we can automatically choose between a random initialization and an initialization from the preceding fast learner with or without an additional behavior cloning regularization term for demonstration. We choose $\tau = 1$ in Proposition 1 across all tasks.

**Sequences of Tasks: MinAtar.** We randomly select 10 sequences of environments and then fix them for reproductivity. We run 3 seeds for each sequence of tasks.

1. ['breakout', 'spaceinvaders', 'breakout', 'spaceinvaders', 'spaceinvaders', 'freeway', 'breakout']

2. ['spaceinvaders', 'breakout', 'breakout', 'spaceinvaders', 'spaceinvaders', 'breakout', 'breakout']

3. ['breakout', 'spaceinvaders', 'breakout', 'freeway', 'freeway', 'breakout', 'freeway']

4. ['freeway', 'breakout', 'spaceinvaders', 'breakout', 'breakout', 'breakout', 'spaceinvaders']

5. ['freeway', 'freeway', 'spaceinvaders', 'spaceinvaders', 'breakout', 'breakout', 'freeway']

6. ['freeway', 'spaceinvaders', 'freeway', 'freeway', 'breakout', 'spaceinvaders', 'breakout']

7. ['freeway', 'spaceinvaders', 'breakout', 'freeway', 'spaceinvaders', 'freeway', 'breakout']

8. ['breakout', 'spaceinvaders', 'freeway', 'breakout', 'spaceinvaders', 'freeway', 'breakout']

9. ['breakout', 'spaceinvaders', 'spaceinvaders', 'spaceinvaders', 'freeway', 'breakout', 'breakout']

10. ['freeway', 'breakout', 'freeway', 'spaceinvaders', 'freeway', 'breakout', 'freeway']

**Remark: Choice of Repeated Tasks.** The shuffled order of tasks is particularly useful to illuminate the mechanism of our adaptive meta warm-up strategy. In real applications, tasks are not strictly one-pass; the newly arriving task is agnostic, which can be either totally unknown or re-encountered with a similar structure—especially in dynamic, cyclical, or seasonal settings. For instance, the house robot is required to clean the floor again in the following weeks after it finishes this week. Although a shuffled order of sequence is slightly less challenging than totally non-repeating scenarios, evaluating mixed types of new tasks is also meaningful for general continual RL scenarios.

**Hyperparameters: Atari Games.** Our implementation adapts from the released code in (Malagon et al., 2024). For our FAME approach based on PPO, we choose the estimation step $N = 20000$, i.e., the number of data to be stored in the meta buffer $\mathcal{M}$ with size 200000 in each task, the policy evaluation step $n = 1200$ for one-vs-all hypothesis test, and the warm-up step $L = 50000$ (5% of the training steps in each task). The ratio of stored data over the whole training data in each task is $20000/1M = 2\%$. In the knowledge integration phase, we train the meta learner across 200 epochs with a $2.5 \times 10^{-4}$ learning rate. The learning rate for the fast learner is also $2.5 \times 10^{-4}$. We adopt $\lambda = 1.0$ for the behavior cloning regularization in the meta warm-up. We choose $\tau = 1$ in Proposition 1 across all tasks.

**Sequences of Tasks: Atari Games.** We follow the setting and adapt the implementation from (Malagon et al., 2024). (1) For the *ALE/SpaceInvaders-v5* environment, we have 10 tasks and each task refers to one playing mode of the game. All tasks share the same objective: shoot space invaders before they reach the Earth. Observations consist of 210 × 160 RGB images of the frames, and actions are: do nothing, fire, move right, move left, a combination of move right and fire, and a combination of move left and fire. The detailed descriptions of the 10 modes are as follows:

- **Mode 0:** It is the default setting. The player has three lives, and destroying space invaders is rewarded (hitting the invaders in the back rows gives more reward).
- **Mode 1:** Shields move back and forth on the screen, instead of staying in a fixed position. Using them as protection becomes unreliable.
- **Mode 2:** The laser bombs dropped by the invaders *zigzag* as they come down the screen, making it more difficult to predict the place they are going to land.
- **Mode 3:** This task combines modes 1 and 2.
- **Mode 4:** Same as mode 0 but laser bombs fall considerably faster.
- **Mode 5:** Same as mode 1 but laser bombs fall considerably faster.
- **Mode 6:** Same as mode 2 but laser bombs fall considerably faster.
- **Mode 7:** Same as mode 3 but laser bombs fall considerably faster.
- **Mode 8:** Same as mode 0 but invaders become invisible for a few frames.
- **Mode 9:** Same as mode 1 but invaders become invisible for a few frames

For the *ALE/Freeway-v5* environment, the objective is to guide a chicken to cross a road with busy traffic. The detailed descriptions of the 7 modes are as follows:

- **Mode 0:** It is the default setting.
- **Mode 1:** Traffic is heavier and the speed of the vehicles increases, the upper lane closest to the center has trucks. Trucks are longer vehicles, and thus, more difficult to avoid.
- **Mode 2:** Trucks move faster than the fastest vehicles of the previous modes, and traffic is heavier.
- **Mode 3:** There are trucks in all lanes, and trucks move as fast as in mode 2 in some of the lanes.
- **Mode 4:** Similar traffic to previous modes, there are no trucks, but the velocity of the vehicles is randomly increased or decreased.
- **Mode 5:** Same as mode 1 with the speed of the vehicles randomly changing and some vehicles come in groups of two or three very close to each other.
- **Mode 6:** Same as the previous mode but with heavier traffic.

### F.2 Ablation Study in MinAtar

**Regularization Hyperparameter $\lambda$ in Behavior Cloning.** Table 4 suggests that an overly large or small $\lambda$ results in inferior performance in FT and Forgetting, although the average performance is still favorable. For example, the metric scores in FT and Forgetting are worst for FAME ($\lambda = 0.1$). In practice, we could choose $\lambda = 1.0$ to achieve the highest score in FT, or $\lambda = 5.0$ for the best forgetting score.

**Warm-Up Step $L$ under the BC Regularization.** Table 5 shows the comprehensive metric scores of FAME in terms of the number of warm-up steps with the BC regularization. It is interesting to highlight that increasing the number of warm-up steps boosts the FT and average performance on

Table 4: Ablation Study of **Regularization Hyperparameter** $\lambda$ on MinAtar on Average Performance (*Avg. Perf*), Forward Transfer (*FT*), and Forgetting. Results are averaged over 10 sequences, each with 3 seeds.

| Method | Ave. Perf ↑ | | | FT ↑ | Forgetting ↓ |
|---|---|---|---|---|---|
| | Breakout | Spaceinvader | Freeway | | |
| FAME ($\lambda$=0.1) | $12.79 \pm 0.42$ | $\mathbf{19.52 \pm 0.50}$ | $\mathbf{1.71 \pm 0.17}$ | $0.13 \pm 0.03$ | $0.77 \pm 0.08$ |
| FAME ($\lambda$=1.0) | $\mathbf{14.54 \pm 0.58}$ | $18.72 \pm 0.52$ | $1.69 \pm 0.17$ | $\mathbf{0.16 \pm 0.03}$ | $0.72 \pm 0.13$ |
| FAME ($\lambda$=5.0) | $13.90 \pm 0.55$ | $19.38 \pm 0.62$ | $1.62 \pm 0.16$ | $0.14 \pm 0.03$ | $\mathbf{0.64 \pm 0.07}$ |
| FAME ($\lambda$=10.0) | $13.88 \pm 0.55$ | $\mathbf{19.52 \pm 0.57}$ | $1.63 \pm 0.16$ | $0.14 \pm 0.03$ | $0.67 \pm 0.08$ |

certain games, such as Spaceinvade and Freeway. However, it worsens the general forgetting. To maintain the forgetting capability, it is recommended to keep the warm-up until a specific phase of the fast learning, such as $5 \times 10^4$ or $20 \times 10^4$ training steps.

Table 5: Ablation Study of **Warm-Up Step** $L$ on MinAtar on Average Performance (*Avg. Perf*), Forward Transfer (*FT*), and Forgetting. Results are averaged over 10 sequences, each with 3 seeds.

| Method | Ave. Perf ↑ | | | FT ↑ | Forgetting ↓ |
|---|---|---|---|---|---|
| | Breakout | Spaceinvader | Freeway | | |
| FAME ($L = 1 \times 10^4$) | $13.34 \pm 0.52$ | $19.17 \pm 0.68$ | $1.64 \pm 0.16$ | $0.13 \pm 0.03$ | $0.74 \pm 0.08$ |
| FAME ($L = 5 \times 10^4$) | $\mathbf{14.54 \pm 0.58}$ | $18.72 \pm 0.52$ | $1.69 \pm 0.17$ | $0.16 \pm 0.03$ | $\mathbf{0.72 \pm 0.13}$ |
| FAME ($L = 20 \times 10^4$) | $13.28 \pm 0.50$ | $19.87 \pm 0.66$ | $1.65 \pm 0.16$ | $\mathbf{0.17 \pm 0.03}$ | $\mathbf{0.72 \pm 0.08}$ |
| FAME ($L = 50 \times 10^4$) | $11.83 \pm 0.71$ | $\mathbf{20.00 \pm 0.96}$ | $\mathbf{1.87 \pm 0.20}$ | $\mathbf{0.17 \pm 0.03}$ | $0.78 \pm 0.09$ |

**Weight Estimation Step** $N$**.** For a fixed size of the meta learner buffer $\mathcal{M}$, we vary the weight estimation step $N$, which determines the collected data in a new environment used for knowledge integration of the meta learner. Table 6 showcases that decreasing the weight estimation step consistently worsens the performance across all metric scores. It is worthwhile to increase $N$ in the future to further enhance our performance, but this would require a larger buffer size of $\mathcal{M}$ than the one employed in other baselines. We leave the investigation of the performance of FAME with a larger buffer size of $\mathcal{M}$ as future work.

Table 6: Ablation Study of **Weight Estimation Step** $N$ on MinAtar on Average Performance (*Avg. Perf*), Forward Transfer (*FT*), and Forgetting. Results are averaged over 10 sequences, each with 3 seeds.

| Method | Ave. Perf ↑ | | | FT ↑ | Forgetting ↓ |
|---|---|---|---|---|---|
| | Breakout | Spaceinvader | Freeway | | |
| FAME ($N$=4000) | $11.95 \pm 0.47$ | $14.69 \pm 0.55$ | $1.54 \pm 0.17$ | $0.14 \pm 0.03$ | $0.93 \pm 0.08$ |
| FAME ($N$=8000) | $12.49 \pm 0.37$ | $17.99 \pm 0.58$ | $1.67 \pm 0.17$ | $0.15 \pm 0.03$ | $0.86 \pm 0.08$ |
| FAME ($N$=12000) | $\mathbf{14.54 \pm 0.58}$ | $\mathbf{18.72 \pm 0.52}$ | $\mathbf{1.69 \pm 0.17}$ | $\mathbf{0.16 \pm 0.03}$ | $\mathbf{0.72 \pm 0.13}$ |

**Policy Evaluation Step** $n$ **for Adaptive Meta Warm-Up before the BC Regularization with** $L$ **Steps.** In Table 7, a proper range of the policy evaluation step does not affect the metric scores significantly. As expected, a small policy evaluation step may not sufficiently select the best warm-up strategy, thus decreasing FT. We found $n = 600$ is sufficient to maintain a favorable FT, while managing the forgetting as well.

Table 7: Ablation Study of **Policy Evaluation Step** $n$ on MinAtar on Average Performance (*Avg. Perf*), Forward Transfer (*FT*), and Forgetting. Results are averaged over 10 sequences, each with 3 seeds.

| Method | Ave. Perf ↑ | | | FT ↑ | Forgetting ↓ |
|---|---|---|---|---|---|
| | Breakout | Spaceinvader | Freeway | | |
| FAME ($n$=300) | $13.02 \pm 0.49$ | $19.19 \pm 0.65$ | $1.42 \pm 0.11$ | $0.12 \pm 0.03$ | $\mathbf{0.69 \pm 0.07}$ |
| FAME ($n$=600) | $\mathbf{14.54 \pm 0.58}$ | $18.72 \pm 0.52$ | $1.69 \pm 0.17$ | $\mathbf{0.16 \pm 0.03}$ | $0.72 \pm 0.13$ |
| FAME ($n$=1200) | $13.46 \pm 0.63$ | $\mathbf{19.39 \pm 0.42}$ | $1.83 \pm 0.20$ | $\mathbf{0.16 \pm 0.03}$ | $0.79 \pm 0.08$ |
| FAME ($n$=5000) | $13.09 \pm 0.41$ | $19.32 \pm 0.52$ | $\mathbf{1.84 \pm 0.18}$ | $0.14 \pm 0.03$ | $0.76 \pm 0.07$ |

### F.3 LEARNING CURVES ANALYSIS FOR PLASTICITY ABILITY

**Learning Curves of the Fast Learner: MinAtar.** We provide the learning curves of all considered continual RL algorithms in the MinAtar environment across 10 sequences of tasks in Figure 4, demonstrating the favorable adaptation capability of the fast learner guided by the adaptive meta warm-up in each new environment.

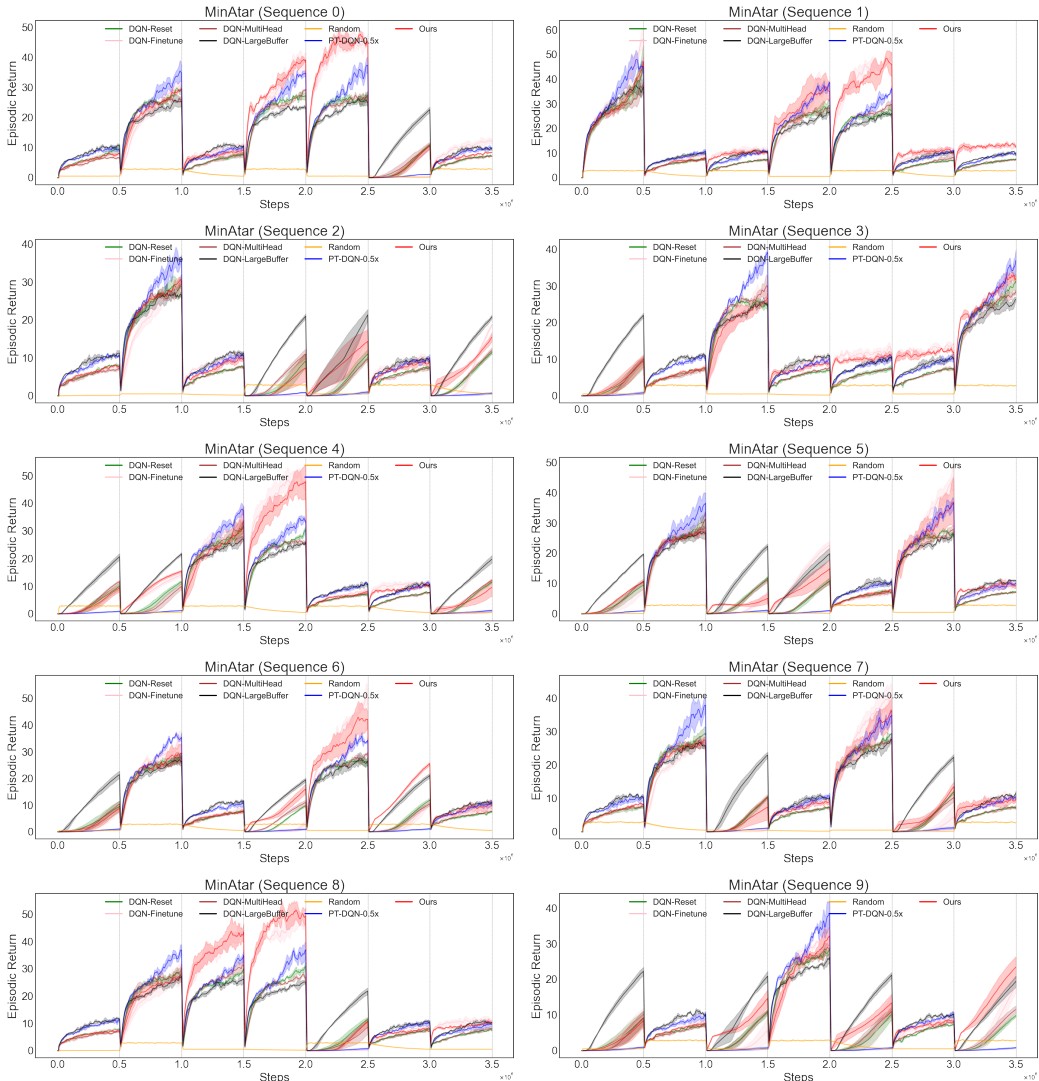

Figure 4: Learning curves of the fast learner in FAME on MinAtar Environments across 10 sequences of tasks.

**Learning Curves of the Fast Learner: Atari Games.** We provide the learning curves of all considered continual RL algorithms in the two Atari environments in Figures 5 and 6. They demonstrate the favorable adaptation capability of the fast learner guided by the adaptive meta warm-up in each new environment.

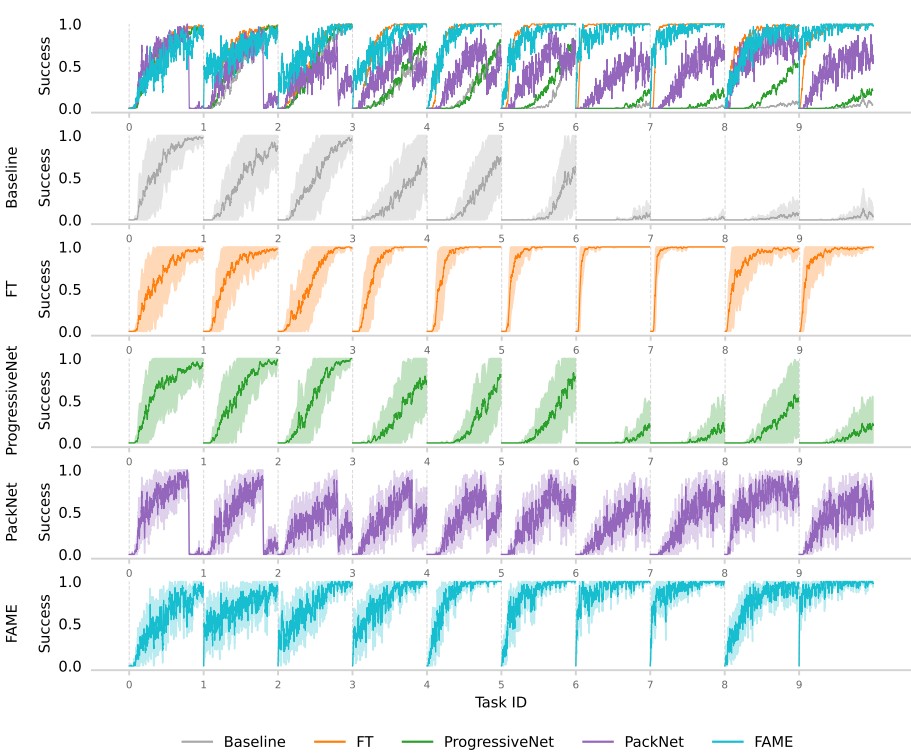

Figure 5: Learning curves of the fast learner in FAME on the SpaceIvader environment averaged over 3 seeds.

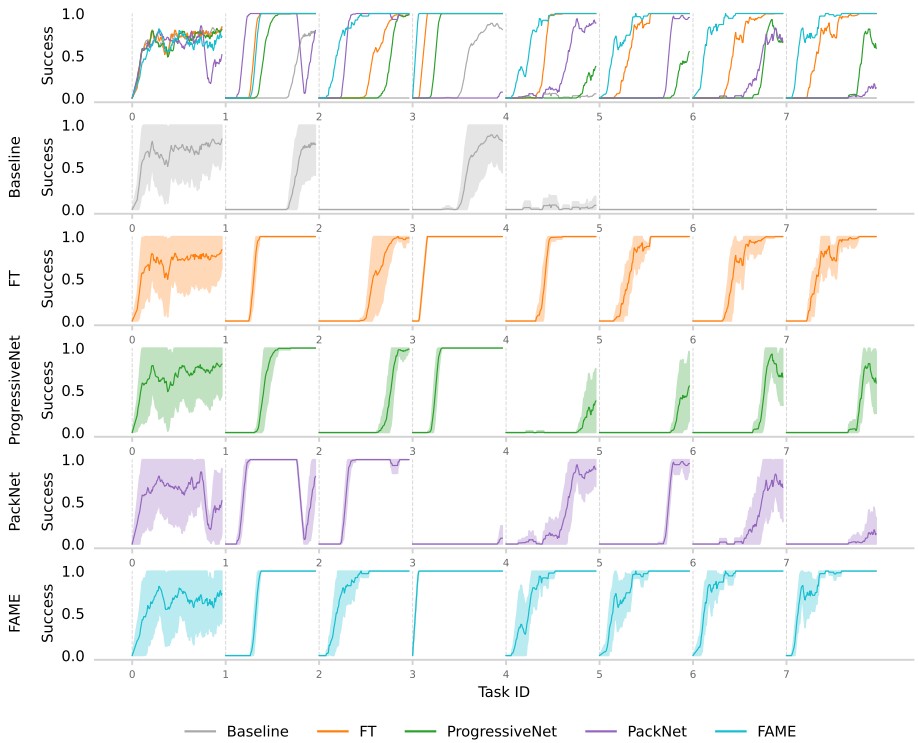

Figure 6: Learning curves of the fast learner in FAME on the Freeway environment averaged over 3 seeds.

# G EXPERIMENTS: POLICY-BASED CONTINUAL RL WITH CONTINUOUS ACTION SPACE

## G.1 DETAILS OF EXPERIMENTAL SETUP AND COMPARISON METHODS

**Hyperparameter Details.** Our implementation adapts from the released code in (Haarnoja et al., 2018b; Chung et al., 2024; Zhang et al., 2024a;b; Malagon et al., 2024). We employ a replay buffer size of $1M$ for the fast learner and $0.1M$ for the meta learner. The buffer size for the meta learner should not be overly large as we expect the algorithm to develop the continual learning capability without replaying too much data in the past. When the new environment arrives, the fast learner starts the training guided by the adaptive meta warm-up (i.e., knowledge transfer). At the same time, the replay buffer for the fast learner is reset, which aims only to contain transitions from the current task. In contrast, the meta learner maintains a buffer that stores the most recent 1% of transitions (50 episodes) leveraged to update the fast learner for each task. After finishing a task, the meta learner is trained using $5000 \times k$ mini-batches, where $k$ is the number of tasks so far.

**Task Sequences.** We randomly choose 3 sequences of tasks from (Chung et al., 2024):

- [`button-press-v2`, `plate-slide-back-side-v2`, `window-close-v2`, `plate-slide-side-v2`, `peg-unplug-side-v2`, `plate-slide-back-v2`, `coffee-button-v2`, `window-open-v2`, `handle-pull-side-v2`, `door-close-v2`],

- [`plate-slide-back-side-v2`, `soccer-v2`, `sweep-into-v2`, `handle-pull-side-v2`, `plate-slide-side-v2`, `peg-unplug-side-v2`, `door-lock-v2`, `reach-v2`, `plate-slide-back-v2`, `coffee-button-v2`],

- [`coffee-push-v2`, `button-press-v2`, `reach-v2`, `peg-unplug-side-v2`, `reach-wall-v2`, `door-close-v2`, `window-open-v2`, `handle-pull-side-v2`, `plate-slide-back-side-v2`, `soccer-v2`].

Table 8: Results on Meta-World on Average Performance (*Ave. Perf*), Forward Transfer (*FT*), and Forgetting **for each sequence of tasks**. Results are presented as averages and standard errors across 10 seeds. Each table, from top to bottom, represents each sequence of tasks, respectively. The best results are highlighted in bold, and the second best are underlined. ↑ denotes a positive metric (more is better), while ↓ is a negative one (less is better). *Reset* is the baseline for evaluating FT.

| Methods | Avg. Perf ↑ | FT ↑ | Forgetting ↓ |
|---|---|---|---|
| Reset | $0.090 \pm 0.029$ | $0.000 \pm 0.000$ | $0.800 \pm 0.040$ |
| Finetune | $0.070 \pm 0.026$ | $-0.294 \pm 0.039$ | $0.480 \pm 0.050$ |
| Average | $0.020 \pm 0.014$ | $-0.584 \pm 0.035$ | $0.100 \pm 0.030$ |
| PackNet | $0.703 \pm 0.041$ | $-0.111 \pm 0.028$ | $\mathbf{0.000 \pm 0.000}$ |
| FAME-WD | $\mathbf{0.87 \pm 0.034}$ | $\underline{0.004 \pm 0.022}$ | $\underline{0.010 \pm 0.017}$ |
| FAME-KL | $\underline{0.86 \pm 0.035}$ | $\mathbf{0.042 \pm 0.019}$ | $0.050 \pm 0.026$ |

| Methods | Avg. Perf ↑ | FT ↑ | Forgetting ↓ |
|---|---|---|---|
| Reset | $0.110 \pm 0.031$ | $0.000 \pm 0.000$ | $0.680 \pm 0.047$ |
| Finetune | $0.040 \pm 0.020$ | $-0.252 \pm 0.045$ | $0.440 \pm 0.052$ |
| Average | $0.000 \pm 0.000$ | $-0.496 \pm 0.039$ | $0.110 \pm 0.031$ |
| PackNet | $0.413 \pm 0.041$ | $-0.249 \pm 0.031$ | $\mathbf{0.000 \pm 0.000}$ |
| FAME-WD | $\mathbf{0.750 \pm 0.044}$ | $\mathbf{0.008 \pm 0.024}$ | $\underline{0.040 \pm 0.032}$ |
| FAME-KL | $\underline{0.680 \pm 0.047}$ | $\underline{0.004 \pm 0.029}$ | $0.120 \pm 0.036$ |

| Methods | Avg. Perf ↑ | FT ↑ | Forgetting ↓ |
|---|---|---|---|
| Reset | $0.080 \pm 0.027$ | $\underline{0.000 \pm 0.000}$ | $0.650 \pm 0.052$ |
| Finetune | $0.000 \pm 0.000$ | $-0.25 \pm 0.036$ | $0.360 \pm 0.048$ |
| Average | $0.020 \pm 0.014$ | $-0.509 \pm 0.036$ | $\underline{0.000 \pm 0.002}$ |
| PackNet | $0.358 \pm 0.043$ | $-0.222 \pm 0.032$ | $\mathbf{0.000 \pm 0.000}$ |
| FAME-WD | $\mathbf{0.680 \pm 0.047}$ | $-0.020 \pm 0.028$ | $0.020 \pm 0.032$ |
| FAME-KL | $\underline{0.660 \pm 0.048}$ | $\mathbf{0.022 \pm 0.028}$ | $0.050 \pm 0.030$ |

Table 9: Results on Meta-World **averaged over the three sequences** on Average Performance (*Ave. Perf*), Forward Transfer (*FT*), and Forgetting. Results are presented as averages and standard errors across 10 seeds. The best results are highlighted in bold, and the second best are underlined. ↑ denotes a positive metric (more is better), while ↓ is a negative one (less is better). *Reset* is the baseline for evaluating FT.

| Methods | Avg. Perf ↑ | FT ↑ | Forgetting ↓ |
|---|---|---|---|
| Reset | $0.093 \pm 0.017$ | $0.000 \pm 0.000$ | $0.710 \pm 0.030$ |
| Finetune | $0.037 \pm 0.011$ | $-0.265 \pm 0.028$ | $0.427 \pm 0.033$ |
| Average | $0.013 \pm 0.007$ | $-0.530 \pm 0.024$ | $0.070 \pm 0.022$ |
| PackNet | $0.491 \pm 0.025$ | $-0.194 \pm 0.018$ | $\mathbf{0.000 \pm 0.000}$ |
| FAME-WD | $\mathbf{0.767 \pm 0.024}$ | $-0.003 \pm 0.014$ | $\underline{0.023 \pm 0.015}$ |
| FAME-KL | $\underline{0.733 \pm 0.026}$ | $\mathbf{0.022 \pm 0.015}$ | $0.073 \pm 0.019$ |

## G.2 MORE EXPERIMENTAL RESULTS AND DETAILS

**Metric Scores.** Table 8 presents detailed results on Average Performance (*Ave. Perf*), Forward Transfer (*FT*), and Forgetting for each sequence of tasks. Table 9 further shows the average score over the three sequences. In summary, compared to the baselines, the FAME variants achieve superior performance in Average Performance and Forward Transfer. For the Forgetting metric: as PackNet retains its previous policies, its Forgetting score is 0. By contrast, Average fails to learn meaningful knowledge and achieves poor overall performance, leaving it with nothing to "forget". As a result, the best Forgetting score does not always align with the best Average Performance.

**Evaluation of Performance Profile and Final Average Performance.** The performance profile (Agarwal et al., 2021; Dolan & Moré, 2002) provides a comprehensive view of the fast learner's overall performance across the entire task sequence. As shown in Figure 7 (left), both FAME variants consistently outperform all baselines, demonstrating the meta-learner's ability to effectively consolidate knowledge over time and facilitate transfer learning. Moreover, Figure 7 (right) highlights the advantage of FAME across all previously seen tasks. While baseline methods typically degrade as more tasks are introduced, FAME-KL and FAME-MD achieve the highest average performance—indicating minimal catastrophic forgetting and robust retention over time. Notably, the last row in Figure 7 indicates the average performance profile and average performance over the three sequences of tasks.

**Learning Curves of the Fast Learner.** Figure 8 showcases that the fast learner in our FAME methods achieves higher success rates across three sequences of tasks throughout training. The first three rows indicate the learning curves for the three sequences of tasks, while the last row represents their average learning curve. Learning curves are averaged over 10 seeds, and the shade region represents the standard error. This superiority implies that the adaptive meta warm-up enhances the adaptation of the fast learner in each new environment.

## H THE USE OF LARGE LANGUAGE MODELS (LLMS)

In this study, large Language Models (LLMs) are only used for minor language polishing and editing. They do not contribute to the theory, methodology, or content development.

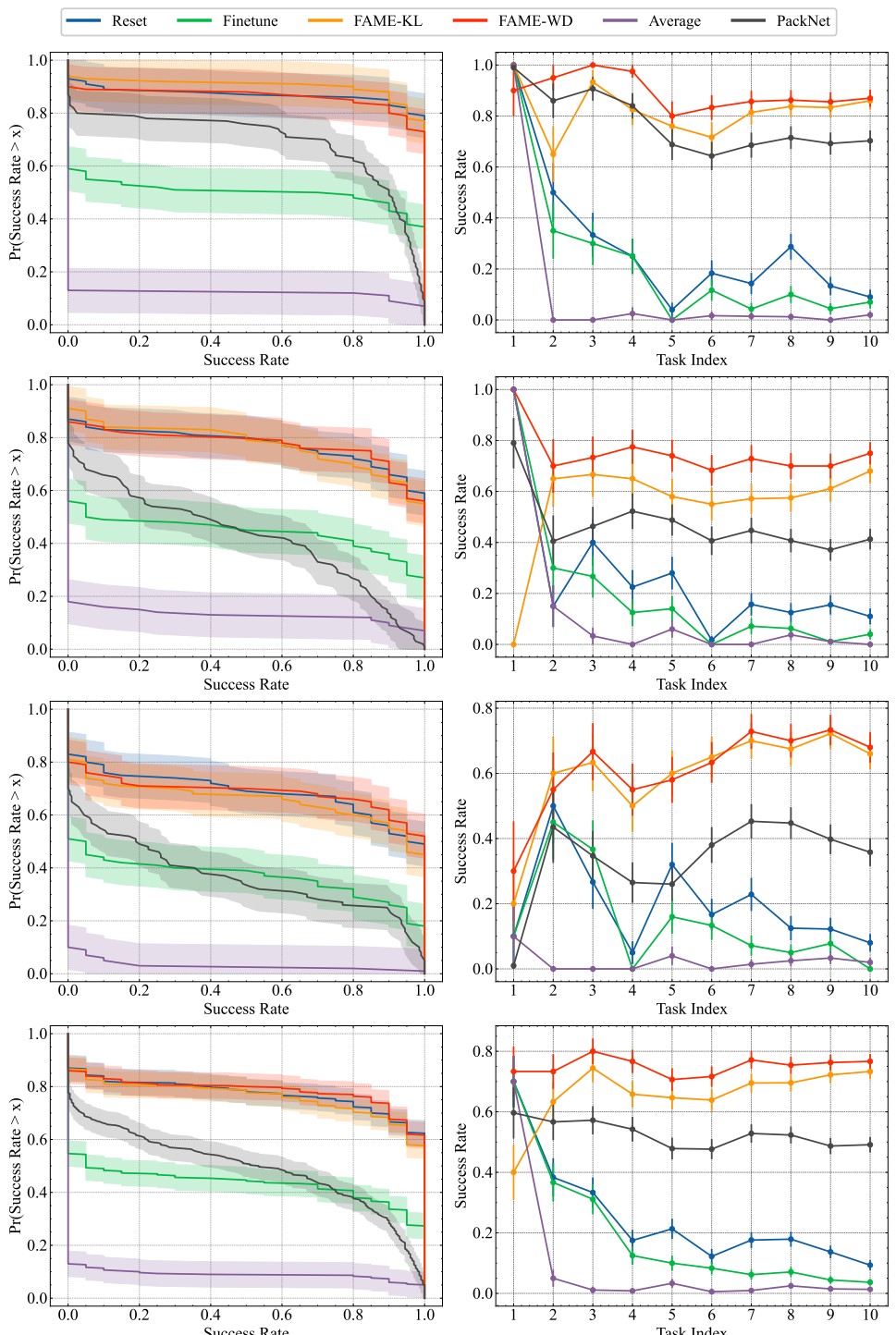

Figure 7: **(Left)** Performance profile of the fast learner across tasks, where the y-axis shows the proportion of tasks that achieve a success rate greater than or equal to the x-axis value. **(Right)** Average performance over time by evaluating the average success rates in the past tasks. **Each Row represents the result for one sequence of tasks. The last row shows the average performance of the three sequences.**

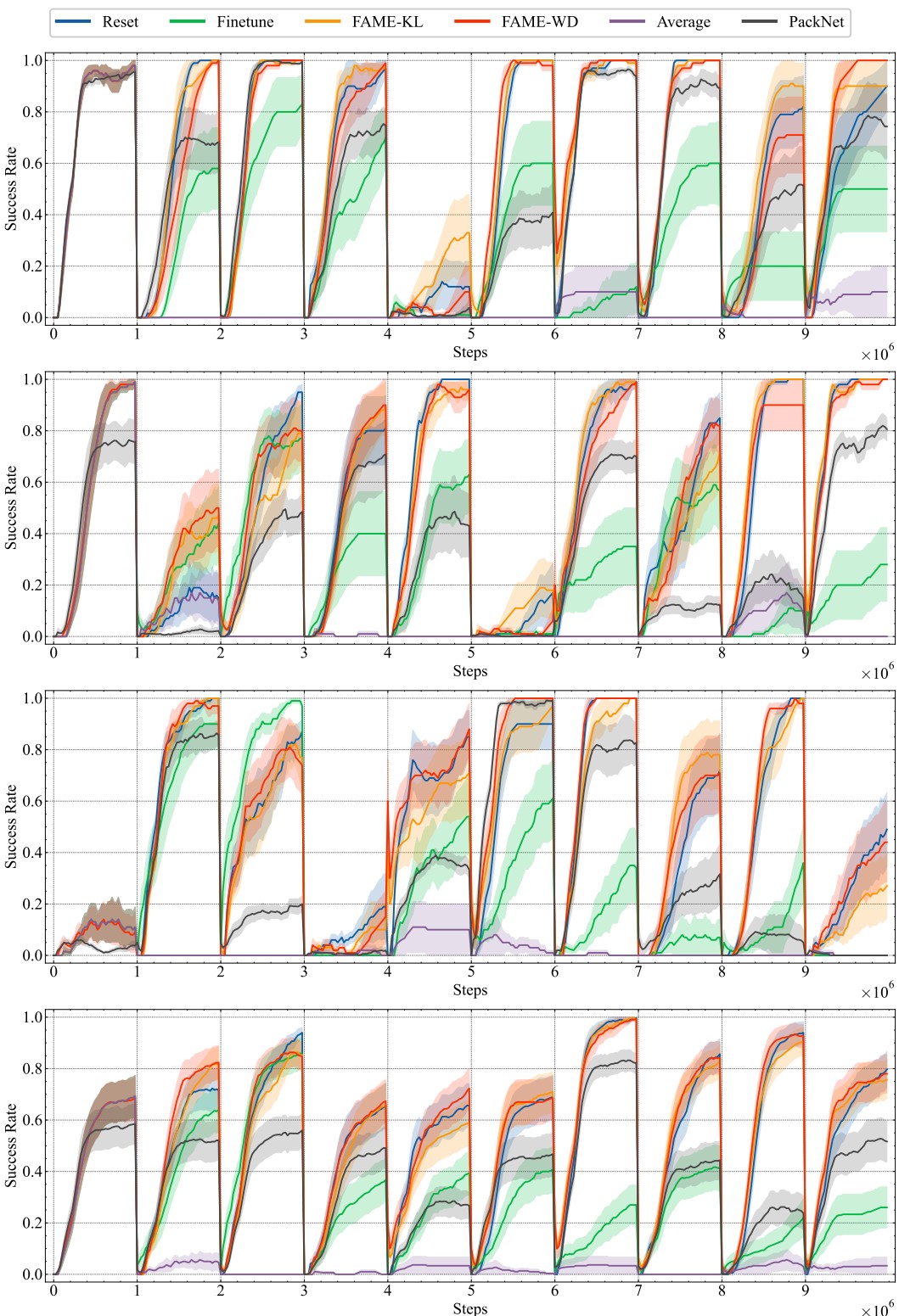

Figure 8: Learning curves of the fast learner on the Meta-World benchmark. The x-axis shows the total number of environment interactions, and the y-axis indicates the success rate. **Each Row represents the result for one sequence of tasks. The last row shows the average performance of the three sequences.**

# I   ADDITIONAL RESULTS

In the camera ready period of this study, we have also added more baselines: ProgressiveNet (Rusu et al., 2016) and CompoNet (Malagon et al., 2024), and a new sequence of task from CW10 in Meta-World environment. The empirical results also suggest the superiority of our FAME algorithms.

## I.1   MORE BASELINES ON THE THREE SEQUENCES OF TASKS IN META-WORLD

Table 10: Results on Meta-World on Average Performance (*Ave. Perf*), Forward Transfer (*FT*), and Forgetting **for each sequence of tasks with more baselines**. Results are presented as averages and standard errors across 10 seeds. Each table, from top to bottom, represents each sequence of tasks, respectively. The best results are highlighted in bold, and the second best are underlined. ↑ denotes a positive metric (more is better), while ↓ is a negative one (less is better). *Reset* is the baseline for evaluating FT.

| Methods | Avg. Perf ↑ | FT ↑ | Forgetting ↓ |
|---|---|---|---|
| Reset | $0.090 \pm 0.029$ | $0.000 \pm 0.000$ | $0.800 \pm 0.040$ |
| Finetune | $0.070 \pm 0.026$ | $-0.294 \pm 0.039$ | $0.480 \pm 0.050$ |
| Average | $0.020 \pm 0.014$ | $-0.584 \pm 0.035$ | $0.100 \pm 0.030$ |
| PackNet | $0.703 \pm 0.041$ | $-0.111 \pm 0.028$ | $\mathbf{0.000 \pm 0.000}$ |
| ProgressiveNet | $0.760 \pm 0.038$ | $-0.063 \pm 0.031$ | $\mathbf{0.000 \pm 0.000}$ |
| CompoNet | $0.803 \pm 0.036$ | $-0.007 \pm 0.022$ | $\mathbf{0.000 \pm 0.000}$ |
| FAME-WD | $\mathbf{0.87 \pm 0.034}$ | $\underline{0.004 \pm 0.022}$ | $\underline{0.010 \pm 0.017}$ |
| FAME-KL | $\underline{0.86 \pm 0.035}$ | $\mathbf{0.042 \pm 0.019}$ | $0.050 \pm 0.026$ |

| Methods | Avg. Perf ↑ | FT ↑ | Forgetting ↓ |
|---|---|---|---|
| Reset | $0.110 \pm 0.031$ | $0.000 \pm 0.000$ | $0.680 \pm 0.047$ |
| Finetune | $0.040 \pm 0.020$ | $-0.252 \pm 0.045$ | $0.440 \pm 0.052$ |
| Average | $0.000 \pm 0.000$ | $-0.496 \pm 0.039$ | $0.110 \pm 0.031$ |
| PackNet | $0.413 \pm 0.041$ | $-0.249 \pm 0.031$ | $\mathbf{0.000 \pm 0.000}$ |
| ProgressiveNet | $0.604 \pm 0.042$ | $-0.133 \pm 0.03$ | $\mathbf{0.000 \pm 0.000}$ |
| CompoNet | $0.566 \pm 0.041$ | $-0.125 \pm 0.027$ | $\mathbf{0.000 \pm 0.000}$ |
| FAME-WD | $\mathbf{0.750 \pm 0.044}$ | $\mathbf{0.008 \pm 0.024}$ | $\underline{0.040 \pm 0.032}$ |
| FAME-KL | $\underline{0.680 \pm 0.047}$ | $\underline{0.004 \pm 0.029}$ | $0.120 \pm 0.036$ |

| Methods | Avg. Perf ↑ | FT ↑ | Forgetting ↓ |
|---|---|---|---|
| Reset | $0.080 \pm 0.027$ | $0.000 \pm 0.000$ | $0.650 \pm 0.052$ |
| Finetune | $0.000 \pm 0.000$ | $\underline{-0.25 \pm 0.036}$ | $0.360 \pm 0.048$ |
| Average | $0.020 \pm 0.014$ | $-0.509 \pm 0.036$ | $0.000 \pm 0.002$ |
| PackNet | $0.358 \pm 0.043$ | $-0.222 \pm 0.032$ | $\mathbf{0.000 \pm 0.000}$ |
| ProgressiveNet | $0.494 \pm 0.044$ | $-0.137 \pm 0.031$ | $\mathbf{0.000 \pm 0.000}$ |
| CompoNet | $0.468 \pm 0.045$ | $-0.136 \pm 0.037$ | $\mathbf{0.000 \pm 0.000}$ |
| FAME-WD | $\mathbf{0.680 \pm 0.047}$ | $-0.020 \pm 0.028$ | $0.020 \pm 0.032$ |
| FAME-KL | $\underline{0.660 \pm 0.048}$ | $\mathbf{0.022 \pm 0.028}$ | $0.050 \pm 0.030$ |

Table 11: Results on Meta-World **averaged over the three sequences with more baselines** on Average Performance (*Ave. Perf*), Forward Transfer (*FT*), and Forgetting. Results are presented as averages and standard errors across 10 seeds. The best results are highlighted in bold, and the second best are underlined. ↑ denotes a positive metric (more is better), while ↓ is a negative one (less is better). *Reset* is the baseline for evaluating FT.

| Methods | Avg. Perf ↑ | FT ↑ | Forgetting ↓ |
|---|---|---|---|
| Reset | $0.093 \pm 0.017$ | $\underline{0.000 \pm 0.000}$ | $0.710 \pm 0.030$ |
| Finetune | $0.037 \pm 0.011$ | $-0.265 \pm 0.028$ | $0.427 \pm 0.033$ |
| Average | $0.013 \pm 0.007$ | $-0.530 \pm 0.024$ | $0.070 \pm 0.022$ |
| PackNet | $0.491 \pm 0.025$ | $-0.194 \pm 0.018$ | $\mathbf{0.000 \pm 0.000}$ |
| ProgressiveNet | $0.619 \pm 0.025$ | $-0.111 \pm 0.016$ | $\mathbf{0.000 \pm 0.000}$ |
| CompoNet | $0.612 \pm 0.025$ | $-0.089 \pm 0.016$ | $\mathbf{0.000 \pm 0.000}$ |
| FAME-WD | $\mathbf{0.767 \pm 0.024}$ | $-0.003 \pm 0.014$ | $\underline{0.023 \pm 0.015}$ |
| FAME-KL | $\underline{0.733 \pm 0.026}$ | $\mathbf{0.022 \pm 0.015}$ | $0.073 \pm 0.019$ |

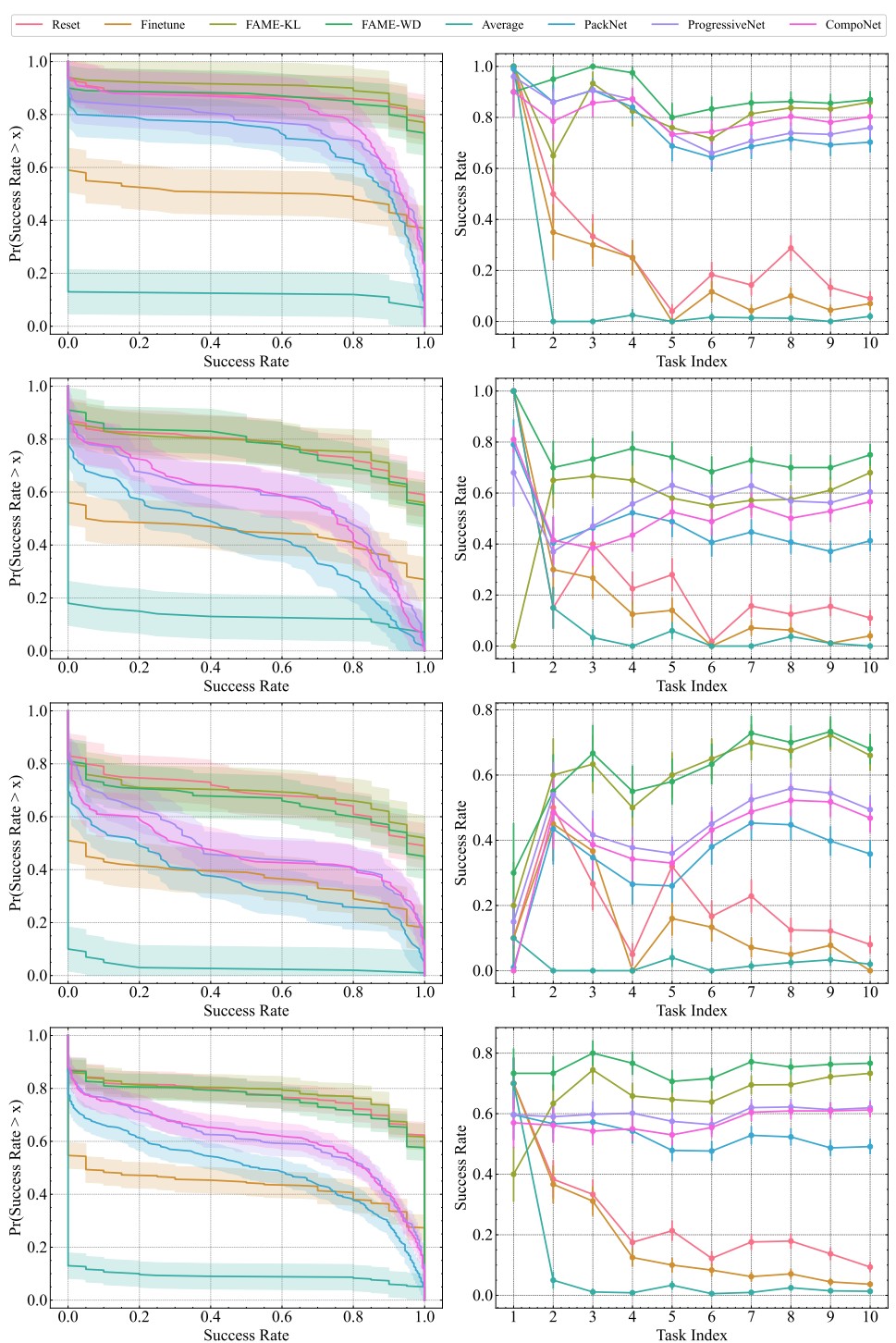

Figure 9: **(Left)** Performance profile of the fast learner across tasks **with more baselines**, where the y-axis shows the proportion of tasks that achieve a success rate greater than or equal to the x-axis value. **(Right)** Average performance over time by evaluating the average success rates in the past tasks. **Each Row represents the result for one sequence of tasks. The last row shows the average performance of the three sequences.**

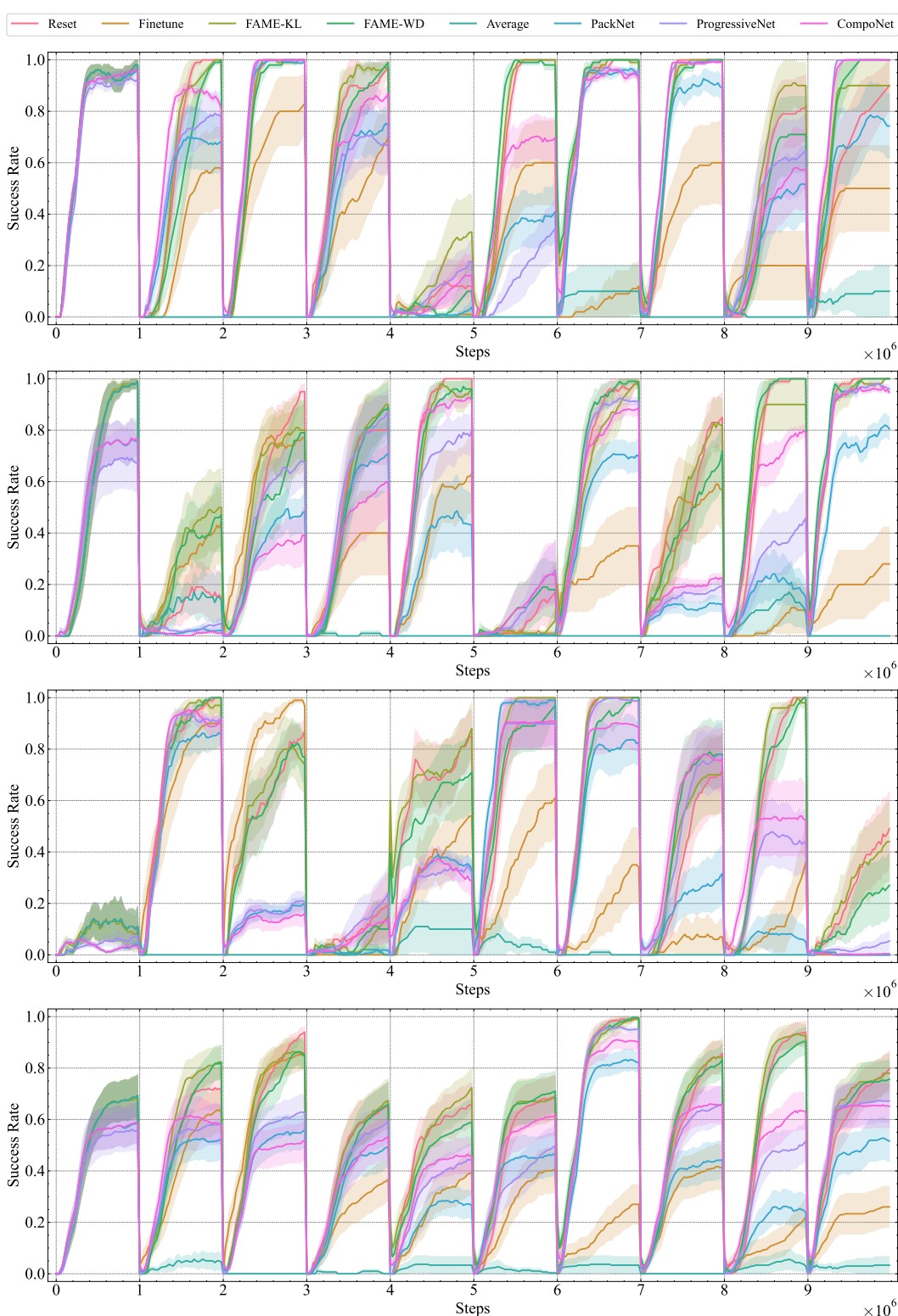

Figure 10: Learning curves of the fast learner on the Meta-World benchmark **with more baselines**. The x-axis shows the total number of environment interactions, and the y-axis indicates the success rate. **Each Row represents the result for one sequence of tasks. The last row shows the average performance of the three sequences.**

## I.2 ADDITIONAL SEQUENCE OF TASK: CW10

Due to time constraints, the baselines of PackNet, ProgressiveNet and CompoNet on CW10 were run with 3 random seeds, while all other baselines were run with 10 random seeds.

Table 12: Results on the sequence of tasks in CW10 (Wolczyk et al., 2021) on Average Performance (*Ave. Perf*), Forward Transfer (*FT*), and Forgetting. Results are presented as averages and standard errors across 10 seeds. The best results are highlighted in bold, and the second best are underlined. ↑ denotes a positive metric (more is better), while ↓ is a negative one (less is better). *Reset* is the baseline for evaluating FT.

| Methods | Avg. Perf ↑ | FT ↑ | Forgetting ↓ |
|---|---|---|---|
| Reset | $0.020 \pm 0.014$ | $\underline{0.000 \pm 0.000}$ | $0.370 \pm 0.049$ |
| Finetune | $0.020 \pm 0.014$ | $-0.123 \pm 0.028$ | $0.240 \pm 0.047$ |
| Average | $0.010 \pm 0.01$ | $-0.288 \pm 0.039$ | $\underline{0.010 \pm 0.017}$ |
| PackNet | $0.292 \pm 0.071$ | $-0.086 \pm 0.019$ | $\mathbf{0.000 \pm 0.000}$ |
| ProgressiveNet | $0.325 \pm 0.074$ | $-0.034 \pm 0.016$ | $\mathbf{0.000 \pm 0.000}$ |
| CompoNet | $0.208 \pm 0.063$ | $-0.129 \pm 0.029$ | $\mathbf{0.000 \pm 0.000}$ |
| FAME-WD | $\underline{0.340 \pm 0.048}$ | $-0.003 \pm 0.016$ | $0.030 \pm 0.017$ |
| FAME-KL | $\mathbf{0.370 \pm 0.049}$ | $\mathbf{0.015 \pm 0.017}$ | $0.060 \pm 0.024$ |

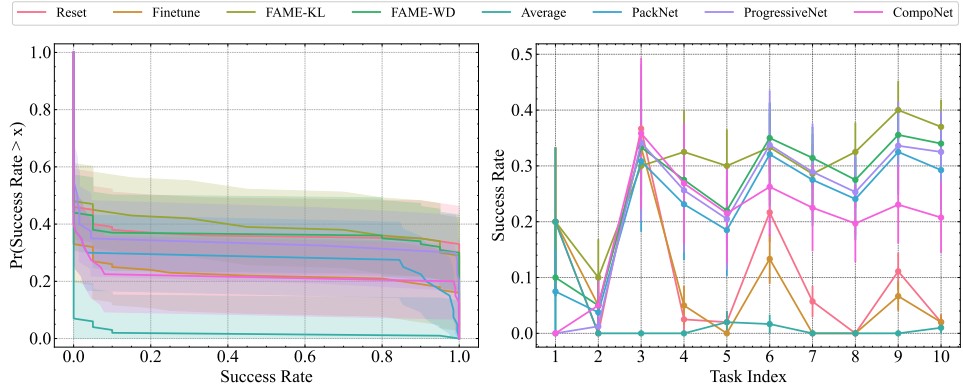

Figure 11: Results on the CW10 (Wolczyk et al., 2021) sequence of tasks. **(Left)** Performance profile of the fast learner across tasks, where the y-axis shows the proportion of tasks that achieve a success rate greater than or equal to the x-axis value. **(Right)** Average performance over time by evaluating the average success rates in the past tasks.

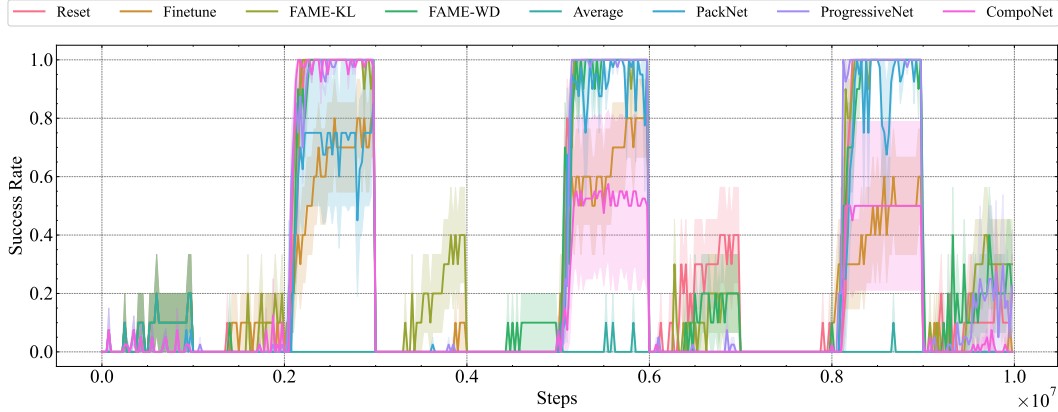

Figure 12: Learning curves of the fast learner on the CW10 (Wolczyk et al., 2021) sequence of tasks. The x-axis shows the total number of environment interactions, and the y-axis indicates the success rate. **Each Row represents the result for one sequence of tasks. The last row shows the average performance of the three sequences.**

