# OpenReview forum: "Principled Fast and Meta Knowledge Learners for Continual Reinforcement Learning"
_ICLR.cc/2026/Conference — ICLR 2026 Poster_

### Official Review · Reviewer_6F7r · 2025-10-19

**Soundness:** 3
**Presentation:** 2
**Contribution:** 3
**Rating:** 4
**Confidence:** 3

**Summary:**

The paper introduces FAME (Principled Fast and Meta Knowledge Learners for Continual Reinforcement Learning), a dual-learner framework designed to address the challenges of continual Reinforcement Learning (RL). The FAME framework decomposes continual RL into two distinct yet complementary objectives handled by the dual-learner system. Fast Learner and Meta Learner.

**Fast Learner (Knowledge Transfer):** This component focuses on rapidly acquiring knowledge from a new task. It leverages an adaptive meta warm-up mechanism that selectively transfers prior knowledge from the meta learner to facilitate rapid adaptation and circumvent the issue of negative transfer. This transfer is achieved either by directly copying parameters or by adding a Behavior Cloning (BC) regularization during the early training phase.

**Meta Learner (Knowledge Integration):** This component ensures knowledge integration and safeguards against catastrophic forgetting. It incrementally integrates new experiences by minimizing catastrophic forgetting, as formally defined in the paper. This consolidation process enhances the adaptive meta warm-up for subsequent environments.

Besides, the authors provide definitions about MDP Distance and Catastrophic Forgetting and provide theoretical results used in FAME.

**Strengths:**

1. Outperforming Performances

2. Clear Motivation

3. Theoretical Foundations

**Weaknesses:**

1. Hyperparameter settings are required, e.g., $L$ and $N$.

2. Inefficiency to train multiple policies

3. Insufficient Abalation Study for $N$ and $L$

**Questions:**

I have two primary inquiries concerning the sensitivity and selection process for the critical hyperparameters $L$ (Warm-Up Step) and $N$ (Weight Estimation Step), whose impact is detailed in the ablation studies.

1.  **On the Non-Monotonic Effect of the Warm-Up Step ($L$):**
    The analysis in Table 5 reveals a complex trade-off associated with the warm-up duration ($L$). While a longer $L$ might intuitively suggest improved learning from the meta-learner, the empirical results show a non-monotonic trend. Specifically, extending $L$ to $50 \times 10^4$ yields marginal gains in Forward Transfer (FT) (0.16 to 0.17) but significantly worsens Forgetting (0.72 to 0.78) and degrades the 'Average Performance' on Breakout. This result is counter-intuitive. Why does a prolonged application of Behavior Cloning (BCI appreciate if you let me know this phenomena.

2.  **On a Principled Strategy for Determining $L$ and $N$:**
    The ablation studies  confirm that performance is highly sensitive to the choice of $L$ and $N$. These parameters appear inherently task-dependent. $N$ dictates the fidelity of the approximated state-visitation distribution ($w_k$ or $\mu_k$), while $L$ manages the plasticity-stability trade-off during adaptation. However, the paper does not seem to offer a principled method for their selection beyond empirical tuning within a specific domain. Could the authors share heuristics or a more adaptive strategy for setting these parameters? For instance, in the Meta-World experiments, the warm-up evaluation was set to 10 episodes. What rationale guided this specific choice, and how might such a heuristic generalize to environments with varying task complexities or episode lengths?

---

> ### Author Response · Authors · 2025-11-19
>
> We would like to sincerely express our gratitude to the reviewer for their time and effort in reviewing our paper. We address each concern you raised and are happy to answer any further questions.
>
>
> > **Weakness: Inefficiency to train multiple policies.**
>
> If we understand it correctly, the reviewer is concerned about the potential computational inefficiency when training both the fast learner and the meta learner. In practice, the additional overhead introduced by the dual-learner framework in FAME is modest.
> * **Space complexity.** The two learners share the same architecture and optimization strategy, so the added memory cost is fixed and minimal.
> * **Computational cost.** For each past task, only a small subset of the training data (approximately 1–2%) is stored in the meta buffer. The increased computational cost is incurred by the meta learner update, which is conducted in a simple supervised-learning way. In MinAtar, for example, updating the meta learner for 200 epochs takes only a few minutes after each task in the GPU server, which is negligible compared to the total computation required for online RL training. Hence, the dual-learner design of FAME remains computationally efficient with only a marginal increase in total cost.
>
> Moreover, compared with model-expansion approaches such as [1, 2, 3], FAME is substantially more efficient. Some of these methods typically require storing and retraining/evaluating all previous policies, whereas FAME maintains only one fast learner and one meta learner. This design makes FAME a principled, elegant, and practical trade-off between computational scalability and continual learning performance.
>
>
>
> > **Question 1. On the Non-Monotonic Effect of the Warm-Up Step $L$.**
>
> Indeed, a longer warm-up duration $L$ in the behavior cloning regularization **does not necessarily** improve overall learning. When the meta learner is selected to warm up the fast learner via the one-vs-all hypothesis test, the conclusion is that the prior knowledge from the meta learner provides a better initialization than either a random learner or the preceding fast learner, and is able to accelerate the training process, especially in the early stages of training. However, as learning progresses, the fast learner gradually adapts to the new environment and can surpass the meta learner’s performance. In such cases, a prolonged behavior-cloning regularization tends to over-constrain the fast learner, thereby hindering adaptation and leading to the observed degradation in final performance and increased forgetting. This explains why we find in Table 5 that there is not a monotonic effect of the warm-up step $L$ and an overly large $L=50\times10^4$ slightly degrades the final performance, such as the average performance on Breakout and the forgetting.
>
> As discussed in the paragraph accompanying Table 5, we recommend using a **moderate** warm-up step $L$, such as $L=5\times10^4$ in MinAtar, instead of using an excessively small or large value. Although the best $L$ is task-dependent, by comparing the results in Tables 1 and 5, the performance of our FAME approach still significantly outperforms the baselines in MinAtar across a broad range of $L$ values. The ablation study in Table 5 of Appendix E.2 further provides practical insights for selecting $L$ in future applications.

---

> > ### Author Response · Authors · 2025-11-19
> >
> > > **Question 2. On a Principled Strategy for Determining $L$ and $N$.**
> >
> > We thank the reviewer for the thoughtful question. As discussed earlier, the optimal warm-up length $L$ is indeed task-dependent. Nevertheless, we observe that using a moderate $L$ already allows FAME to outperform all baselines by a large margin in MinAtar, suggesting that the method is not overly sensitive to precise tuning.
> >
> > Regarding the number of stored trajectories $N$, its selection primarily depends on **computational and memory constraints** (i.e., the meta buffer size). As shown in Table 6 of Appendix E.2, increasing
> > $N$ generally improves performance but comes with higher costs in both memory (increasing the meta buffer size) and computation (meta learner update on a larger trajectory dataset) in the knowledge integration phase. However, updating the meta learner is in a simple supervised learning manner, which is much cheaper than the online RL training.
> >
> > For the Meta-World experiments, the warm-up evaluation was fixed at 10 episodes because this choice already provides strong performance and significantly outperforms other baselines.  Specifically, using a much larger warm-up evaluation step would reduce the number of samples used for online training, as the total number of online interaction steps was kept constant in all experiments for a fair comparison among all methods. Empirically, we found that 10 episodes in the warm-up evaluation already strike a good balance between selecting the most effective prior knowledge and preserving sufficient data for online adaptation.
> >
> > Following your suggestion, we have now added a more detailed discussion about the principled strategy to choose of $L$ and $N$ in **Appendix D**.
> >
> > Thanks again for the reviewer's consistent dedication to reviewing our work, and we are happy to answer any further questions or concerns. Should this rebuttal address your concerns, we would be grateful for an increased score.
> >
> >
> > ### Reference
> >
> > [1] Mikel Malagon, Josu Ceberio, and Jose A Lozano. Self-composing policies for scalable continual reinforcement learning. In Forty-first International Conference on Machine Learning, 2024
> >
> > [2] Jean-Baptiste Gaya, Thang Doan, Lucas Caccia, Laure Soulier, Ludovic Denoyer, and Roberta Raileanu. Building a subspace of policies for scalable continual learning. 2023
> >
> > [3] Arun Mallya and Svetlana Lazebnik. Packnet: Adding multiple tasks to a single network by iterative pruning. In Proceedings of the IEEE conference on Computer Vision and Pattern Recognition, pp. 7765–7773, 2018.

---

> > > ### Comment · Reviewer_6F7r · 2025-11-26
> > >
> > > I thank the authors for their detailed response.
> > >
> > > While the definition of 'moderate steps' remains somewhat empirical, I accept the clarification that the computational costs and parameter tuning burden are manageable in practice. Given the theoretical justification provided alongside the overwhelming empirical performance, I believe my initial concerns are outweighed by the paper's contributions. I acknowledge that I may have initially underestimated the work, and I am raising my score to reflect this.

---

> > > > ### Author Response · Authors · 2025-11-26
> > > >
> > > > We thank the reviewer for the thoughtful follow-up, and we are grateful for the raised score. Thank you for your constructive feedback and for helping improve our work.

---

### Official Review · Reviewer_GXyV · 2025-10-31

**Soundness:** 4
**Presentation:** 3
**Contribution:** 3
**Rating:** 8
**Confidence:** 2

**Summary:**

The paper proposes **FAME**, a principled framework for continual reinforcement learning (CRL) that couples a fast learner (responsible for rapid adaptation and knowledge transfer) with a meta learner (responsible for long-term knowledge consolidation and minimizing catastrophic forgetting). The method is theoretically motivated by defining two new foundations for CRL:
1. MDP Distance, a formal measure of environment similarity.
2. Catastrophic Forgetting, quantified across sequential tasks.

**FAME** integrates these foundations through incremental updates and an adaptive meta warm-up strategy based on hypothesis testing, choosing between finetuning, reset, or meta initialization. Experiments on MinAtar, Atari, and Meta-World benchmarks show strong results across both discrete (DQN-based) and continuous (SAC-based) RL settings.

**Strengths:**

* The paper provides a rigorous definition of MDP similarity and forgetting, which are often treated heuristically in continual RL.
* The dual-learner analogy (hippocampus–cortex interaction) is conceptually elegant and clearly motivates the architectural decomposition.
* The formulation of knowledge integration as minimizing policy-based catastrophic forgetting is mathematically clean and bridges multi-task and continual RL.
* Demonstrates consistent improvement across diverse environments (discrete and continuous), outperforming strong baselines like PackNet, ProgressiveNet, and PT-DQN.

**Weaknesses:**

* The exposition is mathematically dense, and while the theoretical sections are solid, the algorithmic intuition could be clearer (e.g., Eq. 5).
* Although some hyperparameter studies are referenced in Appendix they are not discussed anough in the main paper.
* It’s unclear how the meta buffer scales with very long task sequences or partially observable settings.

**Questions:**

Q: Can you elaborate on the connection between the incremental update rule in Eq. (5) and existing multi-task optimization techniques (e.g., EWC, Distillation, or Distral)?
Q: How does FAME behave when task similarity (MDP distance) is low? Is there a threshold beyond which adaptive warm-up consistently defaults to reset?

---

> ### Author Response · Authors · 2025-11-19
>
> We thank the reviewer for the valuable comments and positive assessment of our work.  We would like to address the concerns you raised in your review.
>
> > **Weakness 1.** The exposition is mathematically dense, and while the theoretical sections are solid, the algorithmic intuition could be clearer (e.g., Eq. 5).
>
> The meta updates, such as Eq.5, is closely connected to multi-task RL. In particular, minimizing the policy-based catastrophic forgetting in Eq. 5 is equivalent to the Maximum Likelihood Estimator (MLE) by fitting the meta learner $Q_k^M$ on a mixture of state-action distributions across all encountered environments.  Similarly, in policy-based continual RL with the usage of KL divergence, the policy-based knowledge integration objective above coincides with the knowledge distillation update used in policy distillation [2] and typical multi-task RL [1], establishing an intriguing connection.
>
> We will also add a short remark in the main text to further clarify the intuition behind each update rule once the paper is accepted and one extra page is available.
>
> > **Weakness 2.** Although some hyperparameter studies are referenced in Appendix they are not discussed anough in the main paper.
>
> Due to the space limit, we defer the hyperparameter studies and detailed experimental setup in Appendix E.1, E.2 and F.1.  We will incorporate a concise discussion of these studies into the main paper to highlight key points in hyperparameter studies and practical settings once the paper is accepted and one extra page is available.
>
>
> > **Weakness 3.** It’s unclear how the meta buffer scales with very long task sequences or partially observable settings.
>
> In FAME, the stored state-action pairs are only a small subset of the training dataset for each task, which is around 1% or 2% in our experiments on the considered benchmarks (i.e., MinAtar, Atari, and Meta-world).  For a fair comparison, we maintain the meta buffer at the same size as the replay buffer used in standard RL, yet FAME consistently achieves superior performance, indicating minimal scalability concerns.
>
> For very long task sequences, one way to scale is by increasing the meta buffer size within the memory budget and dynamically discarding repeated transitions to manage the meta buffer more effectively. Assuming the environment as MDP in our paper is the most common practice, and extending FAME to partially observable settings is an interesting extension, which we consider valuable future work.
>
> > **Question 1.** Can you elaborate on the connection between the incremental update rule in Eq. (5) and existing multi-task optimization techniques (e.g., EWC, Distillation, or Distral)?
>
> The incremental update rule in Eq.(5) indeed resembles the policy distillation and Distral in multi-task RL, but with two key distinctions:
>
> * **Motivation.** Eq.(5) is derived from minimizing the catastrophic forgetting, which is formally quantified by Definition 2 in our paper. While it can be mathematically equivalent to certain multi-task objectives (under softmax policies), our formulation originates from a distinct theoretical principle.
>
> * **Generality.** Our knowledge integration update is not restricted to softmax-based policies. In principle, knowledge integration takes the form of an incremental update, of which Eq. (5) represents a special case. Another important instantiation is Proposition 2, where we use the Wasserstein distance to measure the catastrophic forgetting. That being said, the knowledge integration update defined in our paper can be considered as an incremental update variant of multi-task RL objectives, and can degenerate to the multi-task RL objectives under certain choice of distance metric.
>
> > **Question 2.** How does FAME behave when task similarity (MDP distance) is low? Is there a threshold beyond which adaptive warm-up consistently defaults to reset?
>
> As suggested in Figure 2 (right), our FAME approach tends to select a random initialization (i.e., reset) if the fast learner encounters a new environment with a low task similarity in the task sequence. The threshold is determined automatically through a one-vs-all hypothesis test that compares policy values among the preceding fast learner, the meta learner, and a random policy. If the environment is new compared with the past knowledge, the policy from the preceding fast learner and meta learner tends to have worse performance compared with the reset. In such circumstances, the test favors random initialization, effectively adapting the warm-up mechanism to the degree of task similarity.
>
> ### Reference
>
> [1] Yee Teh, Victor Bapst, Wojciech M Czarnecki, John Quan, James Kirkpatrick, Raia Hadsell, Nicolas Heess, and Razvan Pascanu. Distral: Robust multitask reinforcement learning. NIPS 2017
>
> [2] Andrei A Rusu, Sergio Gomez Colmenarejo, Caglar Gulcehre, Guillaume Desjardins, James Kirkpatrick, Razvan Pascanu, Volodymyr Mnih, Koray Kavukcuoglu, and Raia Hadsell. Policy distillation. arxiv 2015

---

### Official Review · Reviewer_mRHC · 2025-11-01

**Soundness:** 3
**Presentation:** 3
**Contribution:** 3
**Rating:** 6
**Confidence:** 3

**Summary:**

This paper introduces FAME, a biologically inspired framework for Continual Reinforcement Learning that mirrors the complementary roles of the hippocampus and neocortex in human cognition. The framework decomposes continual RL into two coupled processes, where a fast learner that performs rapid adaptation and forward knowledge transfer, and a meta learner that incrementally consolidates knowledge to mitigate catastrophic forgetting. The authors formally define two foundational concepts for continual RL: (1) MDP Distance, which quantifies environment similarity, and (2) a Catastrophic Forgetting metric applicable to both value- and policy-based RL.
On this foundation, FAME employs adaptive meta warm-up and incremental meta updates based on KL and Wasserstein distances to achieve a balance between plasticity and stability. Extensive experiments on pixel-based and continuous control tasks demonstrate consistent improvements in average performance, forward transfer, and resistance to forgetting compared to baselines such as DQN, PPO, PackNet, and ProgressiveNet.

**Strengths:**

1. The paper introduces a principled dual-learner formulation inspired by human memory systems, offering a new perspective on continual RL.
2. The formalization of MDP Distance and Catastrophic Forgetting metrics fills a long-standing theoretical gap, providing measurable quantities for transfer and forgetting analysis.
3. The adaptive meta warm-up and incremental knowledge integration elegantly combine statistical hypothesis testing with RL optimization, resulting in a flexible yet stable continual learning pipeline.

**Weaknesses:**

1. The meta learner’s learned representations are opaque and lack interpretability, and the paper does not analyze the memory or computational scaling of maintaining meta buffers as the number of tasks increases.
2. While theoretically elegant, the algorithm involves multiple nested components, like dual learners, adaptive warm-up, statistical tests, and two types of meta-updates, which may hinder practical adoption or extension without significant engineering effort.

**Questions:**

1. The definition of MDP Distance assumes shared state and action spaces between tasks. How would this metric behave when the state-action distributions of two environments are only partially overlapping or completely distinct?
2. Is there any explicit regularization to prevent the fast learner from overfitting to transient experiences before meta consolidation? If so, how is the trade-off between immediate adaptability and long-term stability controlled?

---

> ### Author Response · Authors · 2025-11-19
>
> We thank the reviewer for the valuable comments and positive assessment of our work.  We would like to address the concerns you raised in your review.
>
> > **Weakness 1.** The meta learner’s learned representations are opaque and lack interpretability, and the paper does not analyze the memory or computational scaling of maintaining meta buffers as the number of tasks increases.
>
> As stated in Line 248 of Section 3.1.3 (Algorithm in Value-based FAME), the stored state-action pairs are only a small subset of the training dataset for each task, which is around 1% or 2% in our experiments on the considered benchmarks (i.e., MinAtar, Atari, and Meta-world).  For a fair comparison, we maintain the meta buffer at the same size as the replay buffer used in standard RL and our FAME approaches still exhibit superior performance across all considered benchmarks.  That being said, our FAME method achieves strong performance with minimal scalability concerns, highlighting its potential to effectively address key challenges in continual reinforcement learning.
>
> Following your suggestion, we have now added a discussion of the scalability issue of storing past trajectories in the meta buffer in **Appendix D** of the revised paper.
>
>
> > **Weakness 2.** While theoretically elegant, the algorithm involves multiple nested components, like dual learners, adaptive warm-up, statistical tests, and two types of meta-updates, which may hinder practical adoption or extension without significant engineering effort.
>
> We would like to clarify that our proposed algorithm is not only theoretically elegant, but also straightforward to implement. Below we summarize the key implementation details for each component:
>
> * **Dual-learners.** From the fast learner's perspective, it integrates the knowledge from the meta learner through a warm-up strategy, either by a behavior cloning regularization or a straightforward initialization. From the meta learner's perspective, the policy is generally updated by integrating the preceding meta learner and the current fast learner, forming an incremental variant of a multi-task objective. Although this dual-learner framework is richer than standard multi-task learning, it builds directly on well-established principles in transfer and multi-task RL, making it conceptually simple to implement.
>
> * **Adaptive warm-up strategy.** In general, adaptive meta warm-up approach chooses the most effective warm-up strategy among the preceding meta learner, a random learner (i.e., reset), and the preceding fast learner (i.e., finetune) according to their policy values. In value-based RL, this is achieved via behavior cloning regularization, while in policy-based RL it corresponds to direct policy initialization.
>
> * **Statistical tests.** The adaptive warm-up employs a one-vs-all hypothesis test, implemented as a standard t-test comparing the mean returns of candidate policies across evaluation episodes. In practice, simply selecting the policy with the highest empirical return ranking often performs comparably, providing an even simpler alternative.
>
> * **Two types of meta-updates.** In value-based continual RL and FAME-KL, the meta update corresponds to Maximum Likelihood Estimation (MLE) under a softmax policy, equivalent to standard multi-task RL [1] and policy distillation [2]. In FAME-WD, the update rule from Proposition 2 involves only the parameterized mean and variance, making it highly efficient and easy to implement.
>
> We have uploaded our code in the submission for your reference. We will release our code to GitHub upon acceptance to facilitate reproducibility and practical adoption.
>
> > **Question 1.** The definition of MDP Distance assumes shared state and action spaces between tasks. How would this metric behave when the state-action distributions of two environments are only partially overlapping or completely distinct?
>
> Theoretically, one can generalize our approach by adopting the Wasserstein distance to measure the divergence between distributions induced by the two optimal policies, as the Wasserstein metric remains well-defined even without shared supports.
>
> Methodologically, a practical extension would be to project environment-specific state–action spaces into a shared latent space, allowing the optimal Q-functions or policies to be compared under a unified representation. This would extend the applicability of the MDP distance to heterogeneous environments. We view this as a promising direction for future work, building upon the current formulation that assumes shared spaces as a foundational step for continual RL.

---

> > ### Author Response · Authors · 2025-11-19
> >
> > > **Question 2.** Is there any explicit regularization to prevent the fast learner from overfitting to transient experiences before meta consolidation? If so, how is the trade-off between immediate adaptability and long-term stability controlled?
> >
> > We would like to clarify that in our framework, the fast learner is intentionally designed to “overfit” to recent experiences for rapid adaptation to a new environment, with support from the meta learner when beneficial (via adaptive meta warm-up). In contrast, the meta learner governs long-term stability by consolidating knowledge and mitigating catastrophic forgetting during the meta update phase. Thus, the adaptability–stability trade-off naturally emerges from the interaction between the fast and meta learners, without requiring additional explicit regularization.
> >
> > ### Reference
> >
> > [1] Yee Teh, Victor Bapst, Wojciech M Czarnecki, John Quan, James Kirkpatrick, Raia Hadsell, Nicolas Heess, and Razvan Pascanu. Distral: Robust multitask reinforcement learning. Advances in neural
> > information processing systems, 30, 2017.
> >
> > [2] Andrei A Rusu, Sergio Gomez Colmenarejo, Caglar Gulcehre, Guillaume Desjardins, James Kirkpatrick, Razvan Pascanu, Volodymyr Mnih, Koray Kavukcuoglu, and Raia Hadsell. Policy distillation. arXiv preprint arXiv:1511.06295, 2015.

---

### Official Review · Reviewer_gasT · 2025-11-01

**Soundness:** 4
**Presentation:** 3
**Contribution:** 3
**Rating:** 8
**Confidence:** 3

**Summary:**

This paper proposes new foundations for continual RL by formally defining measures for catastrophic forgetting and MDP distance—addressing a principle way to understand why continual learning algorithms may or may not work. Building on these foundations, the authors develop Principle Fast and Meta Knowledge RL (FAME), a dual-learner framework that decomposes continual RL into two complementary processes: (1) knowledge transfer via adaptive meta warm-up with hypothesis testing, and (2) knowledge integration via incremental updates that explicitly minimize catastrophic forgetting. Extensive experiments across MinAtar, Atari, and Meta-World validate the approach.

**Strengths:**

- The paper provides the first formal definitions of MDP distance and network level catastrophic forgetting in continual RL, which is directly optimizable.
- The method is sound: It is an adaptive approach considering different knowledge transfer scenarios. It is also flexible, tested on both value-based and policy based algorithms grounded theoretically.
- Experiments are thorough, with diverse domains covering both discrete action spaces (Atari) and continuous action space control (Meta-World). The results shows improvements upon baseline methods.

**Weaknesses:**

- If I understand it correctly, the method requires storing a small sample of trajectories from every tasks, leading to potential scalability issue. The sampling approach is also very simple, with only 1-2% of the training steps, which might cause poor estimation of $\mu$ when the environment gets complex.

**Questions:**

- Regarding the weakness, do you think there is a strategy to discard or store trajectories dynamically from the meta buffer to achieve the same continual learning objective?

---

> ### Author Response · Authors · 2025-11-19
>
> We thank the reviewer for the valuable comments and positive assessment of our work.  We would like to address the concerns you raised in your review.
>
> > **Weaknesses**: If I understand it correctly, the method requires storing a small sample of trajectories from every task, leading to a potential scalability issue. The sampling approach is also very simple, with only 1-2% of the training steps, which might cause poor estimation of $\mu$ when the environment gets complex.
>
> We acknowledge that our method stores a small subset (around 1-2%) of training samples to perform knowledge integration based on the foundations we propose. However, in the considered benchmarks, i.e., MinAtar, Atari, and Meta-world, this strategy does not introduce scalability issues. We use a meta buffer with **the same size** as the replay buffer in standard RL, ensuring fair comparison and manageable resource usage.
>
> We agree that increasing the stored proportion can further improve the estimation and learning in knowledge integration, especially in more complicated environments, which depends on the memory budget. Nevertheless, even with this small fraction, our current approach remains significantly more scalable than many continual RL baselines. For instance, model-expansion methods such as [1, 2, 3] typically require retaining all previous policies or networks to preserve learned knowledge, leading to much higher memory and computational overhead. In contrast, our dual-learner FAME framework achieves strong performance with substantially lower (fixed) storage cost and computational burden.
>
> In summary, FAME provides a competitive and practical continual RL solution with minimal scalability concerns and clear potential for further extension.
>
> > **Question**: Regarding the weakness, do you think there is a strategy to discard or store trajectories dynamically from the meta buffer to achieve the same continual learning objective?
>
> Yes. We agree that developing dynamic storage strategies represents an exciting future direction. One possible improvement is to enhance memory efficiency by detecting and discarding redundant or highly similar state–action pairs in the meta buffer through additional representation-based filtering mechanisms or advanced detection techniques. Another complementary direction is to refine the sampling process during meta experience replay. For instance, it is an interesting direction to adopt a variant of prioritized experience replay [4] speed up the knowledge integration in continual RL by focusing on more informative samples. In summary, we consider the exploration of such dynamic buffer management strategies as promising extensions of the FAME framework.
>
> ### Reference
>
> [1] Mikel Malagon, Josu Ceberio, and Jose A Lozano. Self-composing policies for scalable continual reinforcement learning. In Forty-first International Conference on Machine Learning, 2024
>
> [2] Jean-Baptiste Gaya, Thang Doan, Lucas Caccia, Laure Soulier, Ludovic Denoyer, and Roberta Raileanu. Building a subspace of policies for scalable continual learning. 2023
>
> [3] Arun Mallya and Svetlana Lazebnik. Packnet: Adding multiple tasks to a single network by iterative pruning. In Proceedings of the IEEE conference on Computer Vision and Pattern Recognition, pp. 7765–7773, 2018.
>
> [4] Schaul, T., Quan, J., Antonoglou, I., & Silver, D. (2015). Prioritized experience replay. arXiv preprint arXiv:1511.05952.

---

### Meta-Review · Area_Chair_LvUi · 2026-01-07

**Summary:**

This paper presents a novel continual reinforcement learning (RL) framework based on a dual-learner architecture comprising a fast learner and a meta-learner. It introduces new formalizations for MDP Distance and Catastrophic Forgetting metrics, addressing an open problem in continual RL. The proposed method is technically sound, and empirical results demonstrate its effectiveness across diverse scenarios. Reviewers acknowledged the approach’s novelty, technical rigor, significance, and relevance. Minor concerns regarding memory efficiency and practical implementation were raised but have been adequately addressed in the authors’ rebuttal. Therefore, I recommend acceptance of the paper.

**Reviewer Concerns:**

All reviewers raised concerns about the efficiency of the proposed approach, both in terms of memory and computation. In their rebuttal, the authors adequately addressed these issues, explaining that their method is no less efficient than existing RL approaches. Reviewers mRHC and 6F7r also commented on practical implementation aspects. To address these, the authors provided details on simple heuristics for implementing the method and selecting hyperparameters. Reviewer GXyV noted that the algorithmic intuition was unclear; to resolve this, the authors committed to adding a clarifying remark. Overall, there are no outstanding concerns.

**Reviewer Scores:**

Reviewers gasT and GXyV gave high scores (8) and would likely maintain or possibly increase them after the rebuttal. Reviewer mRHC initially scored the paper 6, and since all their concerns were addressed, they may raise their score. Reviewer 6F7r gave a score of 4 but expressed an intention to increase it following the rebuttal.

---

### Decision · Program_Chairs · 2026-01-26

Accept (Poster)